# Drug-ProGO: A Gene Ontology-Enhanced Contrastive Learning Framework for Drug Virtual Screening with Multi-modality Whole-Protein Input

## Abstract

Virtual screening plays a crucial role in accelerating early-stage drug discovery by efficiently identifying promising small molecule candidates. While most existing methods depend on known binding pockets, protein-level virtual screening has recently gained attention due to its broader applicability in scenarios where pocket information is incomplete or unavailable. In this work, we propose Drug-ProGO, a Gene Ontology (GO) enhanced contrastive learning framework that integrates GO information during training to enrich protein representations, enabling the model to generalize better to unseen proteins by capturing their functional similarity to known ones. This enables the model to better infer compatibility between novel proteins and small molecules. Our framework supports flexible protein inputs, including sequence, structure, and their combination. In the dual-modality setting, the two modalities are processed independently, and their prediction scores are fused using an uncertainty-aware fusion mechanism without additional trainable parameters. Extensive experiments across four virtual screening benchmarks and input settings demonstrate that incorporating GO knowledge consistently improves performance, highlighting the importance of functional knowledge integration for protein-level virtual screening.

## 1 Introduction

Virtual screening plays an essential role in early-stage drug discovery by prioritizing candidate compounds from large chemical libraries (Patel et al., 2021; Schneider, 2010; Sadybekov & Katritch, 2023). Traditional approaches typically rely on molecule docking followed by binding free energy calculations to evaluate binding affinity (Halgren et al., 2004; Trott & Olson, 2010). While being effective, these methods are computationally expensive and scale poorly with the growing size of modern libraries (Zhou et al., 2024). Recent advances in deep learning have led to more efficient alternatives that use neural networks to predict docking conformations and binding affinity, offering a faster and more scalable screening strategy (Zhang et al., 2023b; Cai et al., 2024).

Currently, most deep learning-based virtual screening methods focus on the pocket level (Noor et al., 2024). However, in many practical scenarios, the binding pocket is unknown or unavailable (Zhang et al., 2024). For example, only the protein sequence may be available, yet inhibitors targeting the protein are still needed (Surade & Blundell, 2012). One such case is GPR37L1, an orphan G protein-coupled receptor associated with cardiovascular and neurological disorders (Bolinger et al., 2023). Due to the lack of experimentally resolved structure, structure-based drug discovery for GPR37L1 is particularly difficult. In other situations, the protein structure is known but lacks well-defined or druggable pockets, or the pockets have not been characterized (Jubb et al., 2015). KRAS (Wu et al., 2019), a key oncogenic driver in cancers such as pancreatic and lung cancer, exemplifies this situation. Its relatively flat surface and absence of distinct binding sites have made the development of molecule inhibitors particularly difficult. In such cases, whole-protein level screening emerges as a necessary compromise. Yet, given that mainstream virtual screening, i.e., pocket-based screening, aims to evaluate interactions between candidate molecules and the binding site, the absence of defined pockets makes whole-protein screening considerably more challenging.

When modeling interactions at the whole-protein level, many deep learning approaches rely on supervised training using binding affinity labels (Öztürk et al., 2018; Zheng et al., 2019). However, such high-quality experimental data are scarce and expensive to obtain, which limits the scalability of these methods. Recent contrastive learning frameworks such as DrugCLIP (Gao et al., 2024) offer an alternative by directly aligning protein pocket and ligand features, thereby avoiding the need for explicit affinity labels or docking conformations. Despite this, the number of available protein-ligand pairs remains limited, which poses challenge to model generalization, particularly for proteins without any known ligands.

To more effectively explore the information provided by the limited data of protein-ligand pairs, it is essential not only to learn independent features for each protein but also to capture relationships among proteins. In fact, proteins often have intrinsic connections, such as evolutionary relationships or family classifications, which significantly influence their ligand-binding preferences (Rausell et al., 2010; Kalifa et al., 2024; Zhou et al., 2023b). We observe that Gene Ontology (GO) offers comprehensive and well-structured functional annotations, including molecular functions, biological processes, and cellular components, which can be

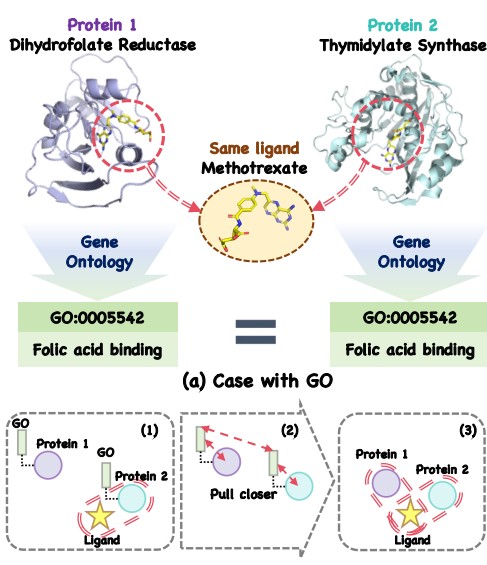

Figure 1: (a) Dihydrofolate Reductase (Protein 1) and Thymidylate Synthase (Protein 2) both bind to the same ligand Methotrexate, but may not appear similar in embedding space. However, both proteins share the Gene Ontology term GO:0005542 (folic acid binding). (b) Without GO supervision (1), Protein 2 is distant from the ligand in the embedding space. By introducing GO-based contrastive learning (2), Protein 1 and Protein 2 are pulled closer via their shared GO term, allowing Protein 2 to align with the ligand (3).

used to explicitly establish these meaningful relationships among proteins (Zhang et al., 2022; Chen et al., 2025). This rich biological context has the potential to alleviate data scarcity. Rather than viewing each protein as an isolated point in the embedding space, GO acts as a bridge connecting proteins and enables the transfer of functional knowledge across them, even for those without experimentally confirmed ligand interactions. Specifically, as shown in Figure 1a, although Dihydrofolate Reductase (Schnell et al., 2004) and Thymidylate Synthase (Costi et al., 2005) both bind Methotrexate (Chan & Cronstein, 2010), their embeddings are far apart due to differences in sequence or structure, making it difficult for at least one of them to be correctly paired with the ligand. However, since both share the GO term (GO:0005542 folic acid binding), incorporating GO information aligns their embeddings in the feature space (Figure 1b). Similarly, ligands known to bind one protein can be implicitly associated with functionally related proteins, enhancing the model's ability to generalize binding relationships. Overall, GO terms not only help mitigate potential conflicts caused by structural or sequential dissimilarities between proteins, but also improve the model's generalization to unseen proteins, thereby enabling more accurate virtual screening in data-scarce scenarios.

However, incorporating GO knowledge into protein representations is challenging, given the hierarchical, redundant, and context-dependent nature of GO terms. This difficulty is amplified in protein-level virtual screening, where protein representations must not only capture functional semantics but also align effectively with ligand features under a contrastive learning framework. To overcome these challenges, we introduce **Drug-ProGO**, a GO-guided protein-ligand contrastive learning framework that integrates GO information into protein encoders. Specifically, Drug-ProGO adopts a contrastive learning framework that aligns protein pocket features with ligand features, encouraging the model not only to distinguish binding from non-binding pairs but also to rank ligands according to their binding affinity labels for the same protein. It further jointly learns from protein-to-GO (P2G) associations and GO-to-GO (G2G) relationships, using both as contrastive signals during training. These terms provide functional links between proteins that go beyond sequence or structure similarity, enabling the model to capture broader biological relationships. During training,

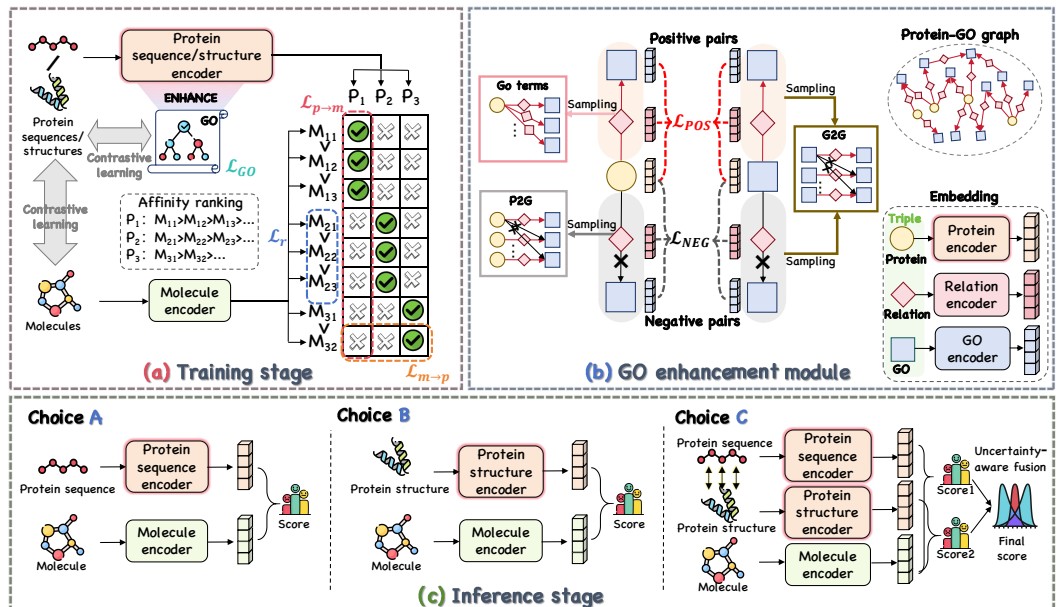

Figure 2: Overview of the Drug-ProGO framework. (a) Training stage: Protein inputs are first encoded using either a sequence encoder or a structure encoder with GO enhancing. In parallel, molecules are processed by a separate encoder. The model is optimized with two contrastive losses to align protein–molecule pairs and support affinity-aware ranking. (b) GO enhancement module: It introduces two types of contrastive supervision: protein-to-GO (P2G) and GO-to-GO (G2G) triples. Positive pairs are drawn from known P2G and G2G relations. For negative sampling, non-associated GO terms are selected from the same ontology aspect. (c) Inference stage: Drug-ProGO supports three inference inputs: sequence (A), structure (B), or both (C). When both modalities are available, an uncertainty-aware fusion mechanism is applied to combine the two predictions.

the model constructs positive and negative pairs across both P2G and G2G views by sampling from curated protein functional annotations and the external GO graph, providing functional supervision to learn biologically meaningful and generalizable representations. However, as GO terms are often unavailable for novel targets in real-world virtual screening, our inference process does not rely on GO, making the method more applicable in practice. As a result, Drug-ProGO demonstrates improved generalization to novel proteins in downstream virtual screening tasks.

Our main contributions are summarized as follows:

1. We introduce GO information into a protein-ligand contrastive learning framework for drug virtual screening. By leveraging the functional relationships encoded in GO, our approach effectively links novel downstream proteins to related proteins seen during training, enabling more accurate prediction of compatible small molecules.

2. We propose a flexible framework that supports whole-protein input with multiple protein representation modalities, including sequence and structure, either separately or combined. Dedicated encoders are trained for each modality, and an uncertainty-aware fusion mechanism is designed to adaptively combine predictions when both modalities are present, enhancing the robustness and reliability of virtual screening results.

3. Drug-ProGO significantly improves performance, achieving state-of-the-art (SOTA) results on four virtual screening benchmarks. **Notably, the incorporation of functional knowledge leads to substantial gains in generalization, especially for unseen proteins.**

4. Drug-ProGO is specifically designed for whole-protein input, enabling its application to drug discovery scenarios where only protein sequences are available or binding pocket structures remain unknown. This approach broadens the scope of virtual screening, demonstrating significant potential in identifying active compounds against novel targets.

## 2 METHOD

Given a target protein $p$ and a set of candidate molecules $\{m_1, m_2, \ldots, m_M\}$, the goal of drug virtual screening is to rank the molecules according to their likelihood of binding to $p$. We propose Drug-ProGO, a Gene Ontology (GO)-enhanced contrastive learning framework that supports multi-modality whole protein inputs and captures semantically enriched protein-molecule alignment.

As shown in Figure 2, Drug-ProGO consists of a molecule encoder and two protein encoders for sequence and 3D structure, respectively. The encoders extract feature representations, and the contrastive objective is used to train the model to increase the similarity between paired protein-molecule embeddings. Furthermore, Drug-ProGO performs ranking among candidate molecules based on their binding affinity labels to a given protein. To enhance functional generalization, GO terms are used to enhance the protein encoders to capture biological semantics and to learn functional relationships between proteins (Figure 2b).

Section 2.1 details the core contrastive learning-based training pipeline (Figure 2a), which aligns protein and molecule features while enabling fine-grained ranking of candidate molecules based on predicted binding affinity. Furthermore, Section 2.2 describes the GO enhancement module (Figure 2b), which constructs contrastive views between proteins and their associated GO terms, as well as among GO terms themselves. Finally, Section 2.3 presents the inference strategy (Figure 2c), where Drug-ProGO supports flexible protein input modalities and integrates an uncertainty-aware fusion mechanism when both sequence and structure are input.

### 2.1 TRAINING STAGE

In this study, we follow the contrastive learning paradigm of LigUnity (Feng et al., 2025) to construct the training stage. For completeness, we briefly introduce its key components below.

Each molecule is represented as a set of atoms characterized by their atom types and 3D coordinates, which are fed into a molecule encoder $E_m$. This encoder is initialized with the pre-trained Uni-Mol (Zhou et al., 2023a) model to leverage prior structural knowledge. The protein is represented at the amino acid level, and its encoding depends on the input modality: if sequence information is available, the protein sequence is encoded by the ESM2 (Lin et al., 2023) model $E_p^{\text{seq}}$; if 3D structure is provided, the protein structure is encoded by the SaProt (Su et al., 2024) encoder $E_p^{\text{struc}}$. Both encoders are further enhanced with functional knowledge via the GO enhancement module. Given the large number of parameters in the pre-trained models, we freeze all parameters of the protein encoder except for the layer norm weights during training.

Through their respective encoders, the protein and molecule inputs are transformed into feature embeddings, denoted as $f_p$ and $f_m$, respectively. These embeddings form the foundation of the training loss, which consists of three components: (1) contrastive loss $\mathcal{L}_c$ for learning protein–molecule compatibility, (2) listwise ranking loss $\mathcal{L}_r$ for capturing fine-grained affinity ranking, and (3) GO-based contrastive loss $\mathcal{L}_{\text{GO}}$ for encouraging functional consistency in protein representations. For more details of $\mathcal{L}_c$ and $\mathcal{L}_r$, please refer to Appendix E.

### 2.2 GO ENHANCEMENT MODULE

To further enrich protein representations with functional semantics, we incorporate GO knowledge into the training process through a contrastive learning mechanism. Specifically, we define two types of knowledge triples: (1) Protein-to-GO (P2G) triples $(p, r, g^+)$, where the head $p$ is a protein, the tail $g^+$ is a GO term, and $r$ denotes the relation type (e.g., *enables*, *located_in*); and (2) GO-to-GO (G2G) triples $(g_h, r, g_t^+)$, where both the head $g_h$ and tail $g_t^+$ are GO terms connected by semantic or hierarchical relations (e.g., *is_a*, *part_of*). These triples serve as structured supervision for learning biologically meaningful embeddings.

To generate positive samples for P2G triples, we randomly select a subset of GO terms associated with each protein. This sampling strategy mitigates overly dense supervision that may occur when proteins are annotated with numerous GO terms. For G2G triples, positive pairs are directly extracted from the GO ontology graph, ensuring that the selected terms reflect true semantic relationships.

Table 1: Results on the DUD-E benchmark. Gray shading indicates the highest-performing method within each category, while bold and underline denote the best and second results across all methods.

| Category | Method | AUROC (%) | BEDROC (%) | EF 0.5% | EF 1% | EF 5% |
|---|---|---|---|---|---|---|
| Traditional | Glide-SP | 76.70 | 40.70 | 19.39 | 16.18 | 7.23 |
| | Vina | 71.60 | – | 9.13 | 7.32 | 4.44 |
| Pocket-input | NN-score | 68.30 | 12.20 | 4.16 | 4.02 | 3.12 |
| | RFscore | 65.21 | 12.41 | 4.90 | 4.52 | 2.98 |
| | Pafnucy | 63.11 | 16.50 | 4.24 | 3.86 | 3.76 |
| | OnionNet | 59.71 | 8.62 | 2.84 | 2.84 | 2.20 |
| | Planet | 71.60 | – | 10.23 | 8.83 | 5.40 |
| | DrugCLIP | 80.93 | 50.52 | 38.07 | 31.89 | 10.66 |
| | LigUnity | 93.10 | 78.86 | **57.14** | **52.04** | 15.77 |
| Protein-input | DeepDTA | 58.36 | 5.13 | – | 2.28 | – |
| | Sequence-DTA | 80.77 | 37.90 | 26.18 | 22.86 | 9.27 |
| | LigUnity-seq | 88.72 | 57.46 | 43.69 | 36.88 | 12.13 |
| | Drug-ProGO(seq) | 93.51 | 74.68 | 53.71 | 48.08 | 15.49 |
| | Drug-ProGO(str) | 93.75 | 76.21 | 54.42 | 49.31 | 15.80 |
| | Drug-ProGO(both) | **94.61** | **79.79** | 56.51 | 51.84 | **16.33** |

Table 2: Results on the LIT-PCBA benchmark.

| Category | Method | AUROC (%) | BEDROC (%) | EF 0.5% | EF 1% | EF 5% |
|---|---|---|---|---|---|---|
| Traditional | Surflex | 51.47 | – | – | 2.50 | – |
| | Glide-SP | 53.15 | 4.00 | 3.17 | 3.41 | 2.01 |
| Pocket-input | Planet | 57.31 | – | 4.64 | 3.87 | 2.43 |
| | Gnina | 60.93 | 5.40 | – | 4.63 | – |
| | BigBind | 60.80 | – | – | 3.82 | – |
| | DrugCLIP | 57.17 | 6.23 | 8.56 | 5.51 | 2.27 |
| | LigUnity | 58.95 | 8.89 | **12.54** | 7.36 | 2.71 |
| Protein-input | DeepDTA | 56.27 | 2.53 | – | 1.47 | – |
| | Sequence-DTA | 55.50 | 3.69 | 2.44 | 2.76 | 2.20 |
| | LigUnity-seq | 56.30 | 7.46 | 11.14 | 6.22 | 2.18 |
| | Drug-ProGO(seq) | 58.13 | 7.91 | 10.77 | 6.65 | 3.05 |
| | Drug-ProGO(str) | 59.27 | 8.34 | 11.65 | 7.32 | 3.13 |
| | Drug-ProGO(both) | **59.89** | **8.98** | 12.40 | **7.65** | **3.26** |

For negative sampling, we employ a GO-guided tail replacement strategy that produces challenging and functionally consistent negative triples. Specifically, for each positive P2G triple $(p, r, g^+)$, the protein $p$ and relation $r$ are kept fixed, while the GO term tail $g^+$ is replaced with a negative GO term $g^-$. To ensure functional consistency and increase the difficulty of discrimination, the negative term $g^-$ is sampled from the same aspect of the GO as $g^+$, but is not associated with the protein $p$. Notably, all GO terms belong to one of three semantic aspects: Molecular Function (MF), Biological Process (BP), or Cellular Component (CC). By sampling within the same GO aspect, we maintain ontological coherence while encouraging the model to learn subtle distinctions between true and false GO terms. Formally, the negative set for P2G triples is defined as:

$$I_{\text{P2G}}^-(p, r_{\text{P2G}}, g^+) = \{(p, r_{\text{P2G}}, g^-) \mid g^- \in G_{\text{aspect}(g^+)}, g^- \notin \mathcal{A}_p\}, \tag{1}$$

where $G_{\text{aspect}(g^+)}$ denotes the collection of GO terms sharing the same aspect as $g^+$, and $\mathcal{A}_p$ is the set of GO terms associated with protein $p$.

Similarly, for G2G triples, negative samples are constructed by corrupting the tail entity. In this case, replacement GO terms are drawn from the same aspect as the original terms but explicitly exclude both the original head and tail terms to avoid trivial negatives. Formally, the negative set for G2G triples is expressed as:

$$I_{\text{G2G}}^-(g_h, r_{\text{G2G}}, g_t^+) = \{(g_h, r_{\text{G2G}}, g_t^-) \mid g_t^- \in G_{\text{aspect}(g_t^+)} \setminus \{g_h, g_t^+\}\}. \tag{2}$$

Notably, P2G and G2G share the same format for loss calculation. Therefore, we use a unified notation to describe them below. We denote the embeddings of the head entity, tail entity, and relation as $f_h$, $f_t$, and $f_r$, respectively. Notably, the GO terms appearing as either head or tail are encoded using a shared GO encoder, initialized from BiomedBERT (Gu et al., 2021), enabling unified representations across both P2G and G2G triples.

Table 3: Results on the DEKOIS 2.0 benchmark.

| Category | Method | AUROC (%) | BEDROC (%) | EF 0.5% | EF 1% | EF 5% |
|---|---|---|---|---|---|---|
| Pocket -input | DrugCLIP | 77.98 | 47.32 | 18.48 | 17.02 | 8.52 |
| | LigUnity | 94.09 | **84.87** | **29.27** | **28.21** | 16.03 |
| Protein -input | Sequence-DTA | 87.85 | 59.34 | 21.17 | 20.23 | 11.69 |
| | LigUnity-seq | 92.46 | 78.48 | 28.06 | 27.13 | 14.30 |
| | Drug-ProGO(seq) | 94.78 | 81.79 | 28.11 | 27.03 | 15.82 |
| | Drug-ProGO(str) | 93.37 | 80.68 | 28.27 | 27.27 | 15.09 |
| | Drug-ProGO(both) | **95.18** | 84.16 | 28.61 | 27.97 | **16.07** |

Table 4: Results on the AD benchmark.

| Category | Method | AUROC (%) | BEDROC (%) | EF 0.5% | EF 1% | EF 5% |
|---|---|---|---|---|---|---|
| Pocket -input | DrugCLIP | 81.19 | 52.04 | 20.50 | 18.00 | 9.10 |
| | LigUnity | 91.33 | 69.51 | 28.15 | 25.36 | 12.14 |
| Protein -input | LigUnity-seq | 90.64 | 71.50 | 28.96 | 25.76 | 12.35 |
| | Drug-ProGO(seq) | 93.56 | 71.30 | 28.93 | 25.67 | 12.95 |
| | Drug-ProGO(str) | 92.90 | 70.24 | 28.23 | 25.18 | 12.84 |
| | Drug-ProGO(both) | **94.26** | **73.67** | **30.12** | **26.72** | **13.34** |

To evaluate the plausibility of triples, we adopt a TransE (Bordes et al., 2013)-inspired scoring function defined by the Euclidean distance: $S_d(f_h, f_t) = \|f_h + f_r - f_t\|_2$.

The GO contrastive loss consists of a positive loss $\mathcal{L}_{\text{POS}}$ and a negative loss $\mathcal{L}_{\text{NEG}}$, and it leverages the scoring function $S_d(f_h, f_t)$ by encouraging positive triples to have smaller distances than negative ones within a margin $\vartheta$. This is formalized as:

$$\mathcal{L}_{\text{GO}} = \mathcal{L}_{\text{POS}} + \mathcal{L}_{\text{NEG}} = -\log \sigma(\vartheta - S_d(f_h, f_t)) - \frac{1}{N} \sum_{n=1}^{N} \log \sigma(S_d(f_{h_n}^-, f_{t_n}^-) - \vartheta), \quad (3)$$

where $\sigma(\cdot)$ is the sigmoid function, $N$ is the number of negative samples, and $(f_{h_n}^-, f_{t_n}^-)$ correspond to the head and tail embeddings of the $n$-th negative triple.

Finally, the total GO contrastive loss, which includes both P2G and G2G components, is integrated into the overall training objective. This integration is done alongside the contrastive loss $\mathcal{L}_c$ and the listwise ranking loss $\mathcal{L}_r$. Therefore, the overall training loss function is defined as: $\mathcal{L} = \mathcal{L}_c + \mathcal{L}_r + \epsilon(\mathcal{L}_{\text{GO}}^{\text{P2G}} + \mathcal{L}_{\text{GO}}^{\text{G2G}})$, where $\epsilon$ is a hyperparameter used to balance the contribution of the GO contrastive loss with the other objectives.

## 2.3 INFERENCE STAGE

As shown in Figure 2c, during inference, the model accepts a protein sequence, a structure, or both as input. It selects the appropriate protein encoder according to the input modality, using either the sequence-based encoder $E_p^{\text{seq}}$ or the structure-based encoder $E_p^{\text{struc}}$ trained during the training stage (Figure 2a). Meanwhile, the molecule is consistently processed by the trained molecular encoder $E_m$.

Given the input protein sequence $x_p^{\text{seq}}$, structure $x_p^{\text{struc}}$, and molecule $x_m$, the encoded features are computed as follows: $f_p^{\text{seq}} = E_p^{\text{seq}}(x_p^{\text{seq}})$, $f_p^{\text{struc}} = E_p^{\text{struc}}(x_p^{\text{struc}})$, $f_m = E_m(x_m)$.

When only one modality is available, the corresponding protein representation is used to compute a similarity score with the molecular embedding. If both sequence and structure inputs are provided, we compute two separate similarity scores: $s_{\text{seq}} = \text{sim}(f_p^{\text{seq}}, f_m), s_{\text{struc}} = \text{sim}(f_p^{\text{struc}}, f_m)$, where $\text{sim}(,)$ typically denotes cosine similarity, measuring the alignment between protein and molecule representations in the shared embedding space.

To effectively combine information from both modalities while mitigating uncertainty from any single source, we adopt an uncertainty-aware rank fusion strategy. This method dynamically adjusts the contributions of the sequence- and structure-based predictions based on their relative uncertainty, measured via entropy.

Table 5: Comparison of Drug-ProGO variants on the DUD-E benchmark. Left: results on unseen proteins. Right: results after retraining on filtered training data. "w/o GO" indicates the corresponding model without GO enhancement. Bold indicates better performance.

| Method | Unseen Proteins | | | | | Retraining | | | | |
|---|---|---|---|---|---|---|---|---|---|---|
| | AUROC | BEDROC | EF 0.5% | EF 1% | EF 5% | AUROC | BEDROC | EF 0.5% | EF 1% | EF 5% |
| w/o GO (seq) | 84.56 | 56.26 | 43.49 | 35.81 | 11.90 | 90.97 | 71.56 | 51.76 | 46.81 | 14.65 |
| Drug-ProGO(seq) | **88.33** | **70.31** | **53.24** | **46.14** | **14.02** | **91.65** | **75.03** | **53.70** | **49.11** | **15.40** |
| w/o GO (str) | 87.95 | 66.57 | 48.42 | 43.53 | 13.78 | 90.86 | 69.41 | 48.80 | 44.68 | 14.78 |
| Drug-ProGO(str) | **89.25** | **68.85** | **50.24** | **44.58** | **14.29** | **91.93** | **71.08** | **50.60** | **45.80** | **14.99** |
| w/o GO (both) | 88.05 | 65.72 | 49.28 | 42.05 | 13.81 | 92.25 | 75.33 | 53.08 | 49.03 | 15.53 |
| Drug-ProGO(both) | **89.62** | **74.19** | **54.79** | **48.36** | **14.80** | **93.01** | **77.45** | **54.40** | **50.34** | **15.94** |

Concretely, we first rank lists $\{s_{\text{seq}}\}$ and $\{s_{\text{struc}}\}$ to $\{R_{\text{seq},i}\}$ and $\{R_{\text{struc},i}\}$, and compute the average rank $\bar{R}_i = (R_{\text{seq},i} + R_{\text{struc},i})/2$ for $i$-th term. We then define inverse-rank probabilities and calculate entropy $H_i$ as:

$$H_i = -\left(P_{\text{seq},i} \log_2 P_{\text{seq},i} + P_{\text{struc},i} \log_2 P_{\text{struc},i}\right), \tag{4}$$

where the probabilities $P$ are defined as:

$$P_{\text{seq},i} = \frac{1/R_{\text{seq},i}}{1/R_{\text{seq},i} + 1/R_{\text{struc},i}}, P_{\text{struc},i} = \frac{1/R_{\text{struc},i}}{1/R_{\text{seq},i} + 1/R_{\text{struc},i}}. \tag{5}$$

The final fused ranking score is computed by weighting the average rank with the entropy-derived uncertainty: $s_i^{\text{fused}} = -\bar{R}_i \cdot \exp(\lambda H_i)$, where $\lambda$ is a hyperparameter controlling the sensitivity to uncertainty. The negative sign ensures that higher scores correspond to better ranking.

This uncertainty-aware fusion mechanism allows the model to dynamically weigh structural and sequential representations based on their estimated reliability. Instead of treating both modalities equally, it assigns greater importance to the modality with lower uncertainty, i.e., lower entropy, thereby reducing the risk of performance degradation due to noisy or misleading inputs.

## 3 EXPERIMENTS AND RESULTS

To evaluate the effectiveness of Drug-ProGO, we assess its performance under three whole-protein input settings: using sequence information only ("seq"), structure information only ("str"), and a combination of both ("both"). Experiments are conducted on four widely adopted virtual screening benchmarks: DUD-E (Mysinger et al., 2012), LIT-PCBA (Tran-Nguyen et al., 2020), DEKOIS 2.0 (Bauer et al., 2013), and AD (Chen et al., 2019). Each benchmark comprises multiple protein targets, where each target is associated with a set of known active ligands as well as a corresponding set of decoys. The goal is to rank candidate molecules for each protein target such that active compounds appear as early as possible in the ranked list. More details of benchmarks can be obtained in Appendix C. The performance comparisons with existing methods under each input setting are presented in section 3.1. We follow previous studies and evaluate screening performance using AUROC, BEDROC, and enrichment factor (EF), as detailed in Appendix F.

For training, we utilize the training dataset provided by LigUnity (Feng et al., 2025), which contains protein-ligand binding pairs together with binding affinity values measured under consistent experimental conditions and units. This uniformity ensures that affinity-based rankings are directly comparable across samples. LigUnity-seq uses 12-layer, 35M parameter ESM-2 model, whereas our sequence encoder employs 33-layer, 650M parameter ESM-2 model. Additional details regarding training dataset and hyperparameter settings are available in the Appendix B.

Moreover, we evaluate the model's generalization ability to unseen proteins in Section 3.2. Following that, a comprehensive ablation study is presented in Section 3.3 to quantify the contributions of individual components. Finally, the case study can be obtained in Section 3.4.

## 3.1 PERFORMANCE ON VIRTUAL SCREENING BENCHMARKS

**DUD-E benchmark**: As shown in Table 1, we compare our method with three representative categories: traditional computational methods (Glide-SP (Halgren et al., 2004), Vina (Trott & Olson, 2010)), deep learning models that rely solely on pocket input (NN-score (Durrant & McCammon, 2011), RFscore (Ballester & Mitchell, 2010), Pafnucy (Stepniewska-Dziubinska et al., 2018), OnionNet (Zheng et al., 2019), Planet (Zhang et al., 2023a), DrugCLIP (Gao et al., 2024), LigUnity (Feng et al., 2025)), and those that take entire protein as input (DeepDTA (Öztürk et al., 2018), Sequence-DTA (Feng et al., 2025), LigUnity-seq (Feng et al., 2025)). Pocket-input methods are known to benefit from explicit knowledge of the binding region, which helps clarify the interaction between the molecule and the specific protein site, thereby enhancing virtual screening performance. In contrast, whole protein-input models face a more difficult scenario, as they must infer interaction sites without prior structural guidance, where all three variants ("seq", "str", "both") of Drug-ProGO fall into the protein-level input category and consistently achieve SOTA results. Specifically, "both" exceeds "str", and "str" surpasses "seq", which confirms the significance of structure input and supports the robustness of our uncertainty-aware fusion mechanism. Notably, Drug-ProGO(both) outperforms LigUnity, the current SOTA method in pocket-input category, across most evaluation metrics, highlighting the superior capability of Drug-ProGO in virtual screening.

**LIT-PCBA benchmark**: Table 2 presents the performance comparison between three variants of Drug-ProGO and a set of baseline methods. Among the baselines, Surflex (Spitzer & Jain, 2012) and Glide-SP (Halgren et al., 2004) represent traditional computational methods, while others are deep learning models based on either pocket-level (Planet (Zhang et al., 2023a), Gnina (McNutt et al., 2021), Big-Bind (Brocidiacono et al., 2023), Drug-CLIP (Gao et al., 2024), LigUnity (Feng et al., 2025)) or protein-level inputs (Deep-DTA (Öztürk et al., 2018), Sequence-DTA (Feng et al., 2025), LigUnity-seq (Feng et al., 2025)). Under whole-protein input

Table 6: Ablation studies on GO enhancement (upper) and uncertainty-aware fusion (lower).

| Method | AUROC (%) | BEDROC (%) | EF 0.5% | EF 1% | EF 5% |
|---|---|---|---|---|---|
| w/o GO (seq) | 92.02 | 68.98 | 48.85 | 43.91 | 14.76 |
| Drug-ProGO(seq) | **93.51** | **74.68** | **53.71** | **48.08** | **15.49** |
| w/o GO (str) | 93.12 | 72.34 | 51.92 | 46.86 | 15.12 |
| Drug-ProGO(str) | **93.75** | **76.21** | **54.42** | **49.31** | **15.80** |
| w/o GO (both) | 93.67 | 75.70 | 53.20 | 48.77 | 15.87 |
| Drug-ProGO(both) | **94.61** | **79.79** | **56.51** | **51.84** | **16.33** |

| Method | AUROC (%) | BEDROC (%) | EF 0.5% | EF 1% | EF 5% |
|---|---|---|---|---|---|
| Add | 94.45 | 79.08 | 56.45 | 51.61 | 16.15 |
| Learned head | 93.31 | 74.14 | 53.26 | 47.74 | 15.43 |
| Drug-ProGO(both) | **94.61** | **79.79** | **56.51** | **51.84** | **16.33** |

settings, Drug-ProGO consistently achieves SOTA results across all evaluation metrics. The superior results of "both" variant, followed by structure and then sequence, indicate that structural information contributes more substantially to model performance and that our uncertainty-aware fusion mechanism effectively leverages multi-modality data. In particular, the "both" variant, which incorporates both sequence and structure information, performs competitively with leading pocket-based deep learning methods. This demonstrates the effectiveness of the GO-enhanced protein encoder.

**DEKOIS 2.0 benchmark**: As shown in Table 3, Drug-ProGO achieves consistently strong performance across all input modalities and evaluation metrics. Notably, the "both" variant of our method performs on par with advanced pocket-based models, despite not relying on pocket information. These results validate the generalization ability of our GO-enhanced encoder.

**AD benchmark**: As shown in Table 4, Drug-ProGO consistently delivers strong performance across different input modalities and evaluation metrics. Even without access to pocket information, Drug-ProGO(both) achieves superior performance compared to the existing pocket-input SOTA method, underscoring the advantage of integrating GO information.

## 3.2 EVALUATION ON UNSEEN PROTEIN

To evaluate the generalization capability of our method, we exclude all target proteins of the DUD-E benchmark that appear in training dataset for further experiment. We then compare the performance of our method and its counterpart without GO enhancement on these unseen targets. As shown in Table 5(left), the GO-enhanced model significantly outperforms the version without GO enhancing across all metrics, with particularly notable gains on the BEDROC metric, where performance improves by +14.05% in sequence input. This highlights the substantial benefit of GO enhancement when encountering previously unseen proteins.

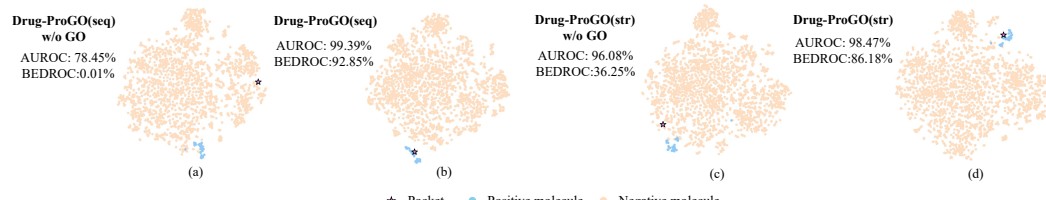

Figure 3: T-SNE feature visualization on the SAHH target. Drug-ProGO(seq) (b) and Drug-ProGO(str) (d) are compared with their respective variants without GO enhancement, i.e., Drug-ProGO(seq) w/o GO (a) and Drug-ProGO(str) w/o GO (c).

Furthermore, we remove from our original training dataset all proteins that appear in the DUD-E benchmark to ensure strict evaluation on novel proteins. Drug-ProGO is retrained on this filtered data. As shown in Table 5(right), all three variants of Drug-ProGO consistently outperform their respective counterparts without GO enhancement. The improvement is particularly pronounced in the sequence-only setting, highlighting the functional information in compensating for the absence of structural cues. Additional similarity clustering experiments are reported in Appendix H.

## 3.3 ABLATION STUDY

We conduct all ablation studies on the DUD-E benchmark. As shown in Table 6(upper), removing GO information leads to a significant performance drop across all input modalities. The effect is particularly significant in the sequence-only setting, where our method leads to +5.70% in BEDROC.

To further validate the effectiveness of our uncertainty-aware fusion mechanism, we compare it with two baselines. The first is a simple addition of Drug-ProGO(seq) and Drug-ProGO(str) outputs, while the second is a learnable fusion that freezes both encoders and trains a small MLP head with ranking loss to combine their scores. As shown in Table 6 (lower), our parameter-free, uncertainty-aware fusion consistently outperforms both alternatives. The learnable fusion does not improve performance, likely due to mild distribution mismatches between sequence and structure encoders, which can lead a trainable head to overfit. These results justify our choice of a non-parametric fusion design.

Furthermore, we conduct an ablation study on the number of sampled GO terms for each protein and the number of sampled negative triples. We evaluate four settings with 1, 3, 5, and 7 entries, respectively. As shown in Figure 7, the setting with five GO terms and three negative samples achieves the best performance. More ablation results are provided in Appendix G.

## 3.4 VISUALIZATION

As illustrated in Figure 3, we visualize the feature distributions of Drug-ProGO under both sequence-only and structure-only inputs on the unseen target SAHH from the DUD-E benchmark. Without GO enhancement, although the active features appear tightly clustered, they are significantly deviated from the distribution of the target-specific features, suggesting a poor alignment with the target. In contrast, after introducing GO enhancement module, the distributions of active molecules' features are better aligned with the target features, highlighting the necessity and effectiveness of enhancing the protein encoder using GO-based functional knowledge. More case study refer to Appendix K.

To further analyze how the GO information improves the model's performance, we visualize the distance between each DUD-E target and its nearest training protein embedding, comparing models trained on unseen proteins with and without GO enhancement for both sequence- and structure-inputs. As shown in Figure 4, introducing GO consistently reduces these distances, indicating that unseen test proteins become closer to the training distribution. This suggests that GO acts as a functional bridge that reshapes the representation space, allowing proteins with low sequence or structural similarity to be aligned with biologically relevant neighbors and thereby improving generalization in virtual screening.

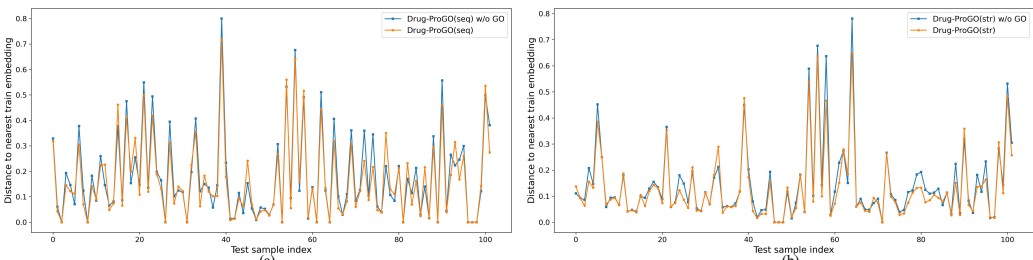

Figure 4: Visualization of how GO enhancement affects the embedding distance between DUD-E test proteins and the nearest training protein. (a) Sequence-based embeddings. (b) Structure-based embeddings.

## 4 CONCLUSION

In this work, we propose a virtual screening approach tailored for whole-protein inputs, particularly suited for scenarios where only the protein sequence is available or the binding pocket is undefined. To improve generalization to unseen proteins, we introduce a GO-enhanced training strategy that incorporates GO information exclusively during the training phase. By leveraging contrastive learning, our method encourages the model to capture biological relationships among training proteins, rather than relying solely on local similarities between individual protein-molecule pairs. Notably, the trained model operates without GO information during inference, enabling its application to novel targets with unknown or incomplete biological annotations. Overall, our method provides an effective solution for pocket-free virtual screening and holds strong potential for real-world drug discovery involving unseen proteins.

## ETHICS STATEMENT

This study does not involve experiments on human participants, the use of sensitive personal information, or applications with dual-use risks. The datasets utilized are entirely public and have been widely adopted in prior literature, with proper licensing and attribution. No confidential or proprietary data are used, and the proposed methods are developed exclusively for advancing research in computational biology and machine learning. The work does not raise ethical concerns regarding fairness, bias, privacy, or security.

## REPRODUCIBILITY STATEMENT

Comprehensive information on the model design, hyperparameter settings, training procedures, and inference steps is provided in the main text and the Appendix. All experimentals are carried out on publicly accessible datasets, and we commit to releasing the implementation and trained models to facilitate complete reproducibility of our results.

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

## A  RELATED WORK

Virtual screening (VS) plays a pivotal role in early-stage drug discovery by efficiently identifying candidate molecules that bind to target proteins ref1,ref2. Existing VS approaches can be broadly categorized into traditional docking-based methods and emerging deep learning-based techniques(Gao et al., 2024).

Traditional docking-based methods, such as AutoDock Vina (Trott & Olson, 2010), Glide (Halgren et al., 2004), and Surflex (Spitzer & Jain, 2012), rely on a predefined binding pocket, typically specified as a docking box. These methods explore favorable ligand conformations within the designated region and evaluate binding using physics-based scoring functions. As a result, their performance is highly sensitive to the accuracy of pocket definition and docking poses. However, in real-world applications, pocket information is often unavailable, unreliable, or requires manual specification. This significantly limits the applicability of these methods to targets that lack experimentally resolved structures or clearly defined binding sites.

In contrast, deep learning has recently emerged as a promising alternative. Most existing deep learning-based VS methods concentrate on pocket-level representations. A major class of these methods follows a supervised learning paradigm. Some models, such as CarsiDock (Cai et al., 2024) and KarmaDock (Zhang et al., 2023b), first predict protein-ligand binding poses and then score the resulting complexes. Others, including OnionNet (Zheng et al., 2019) and Planet (Zhang et al., 2023a), directly predict binding affinities based on complex structures and experimental measurements. Protein-level methods such as DeepDTA (Öztürk et al., 2018) and GraphDTA (Nguyen et al., 2021) also belong to this category. Although these approaches have shown good performance in benchmark datasets, they generally depend on high-quality labeled data, such as accurate docking poses or experimentally measured affinities. Such requirements are often unmet in realistic virtual screening scenarios, where the goal is to identify active compounds from a large set of unlabeled candidates.

To address these limitations, another line of research adopts contrastive learning frameworks. Methods such as DrugCLIP (Gao et al., 2024) learn to score and rank molecules by measuring the similarity between protein and ligand representations, without relying on precise binding poses or explicit affinity values (Feng et al., 2025). These methods are more consistent with the objectives of practical virtual screening, particularly in settings where structural annotations or bioactivity labels are unavailable. Motivated by this line of work, our method also builds upon the contrastive learning paradigm to better support large scale protein-level virtual screening.

## B  DETAILS OF TRAINING DATASET

Our training dataset is based on the dataset used in the LigUnity (Feng et al., 2025) and consists of two main components. The first component includes 26,748 entries collected from ChEMBL (Zdrazil et al., 2024) and BindingDB (Gilson et al., 2016). Each entry originates from a single biological assay, ensuring that ligand affinity measurements are obtained under consistent experimental conditions and reported using uniform units ($K_i$, $K_d$, or $IC_{50}$). This subset comprises 4,847 unique proteins and 53,406 unique molecules. The second component is derived from the PDBBind (Wang et al., 2005), containing 16,744 protein–ligand pairs. Since PDBBind does not provide detailed information about experimental conditions, we treat each protein–ligand pair as an independent data point, resulting in 2,196 unique proteins and 428,767 unique ligands. As in LigUnity, interactions from ChEMBL and from BindingDB entries that share the same assay are supervised with the ranking loss, whereas PDBBind samples without assay-consistent affinity labels are treated as unlabeled and trained purely through contrastive objectives.

For each protein, we retrieve GO terms using its UniProt ID from the QuickGO (Binns et al., 2009). We then map both GO terms and their associated relations according to the encoding scheme provided by OntoProtein (Zhang et al., 2022). All UniProt IDs are enumerated, and each GO term triple is encoded as a P2G triple: (protein ID, relation ID, GO term ID). A total of 110,146 G2G triples are obtained from OntoProtein (Zhang et al., 2022) and are uniformly assigned to each entry during training. During model input construction, each component is embedded using its corresponding encoder.

We analyze the number of GO terms per protein entry. As shown in Figure 5, most proteins are annotated with no more than 5 GO terms. In particular, 2,066 entries contain 5 or fewer annotations. Based on this observation, we limit the number of sampled GO terms per protein to 5. This design ensures that the majority of proteins have sufficient functional context while minimizing sparsity.

Additionally, we examine the count distribution of GO relation types across the entire dataset. A total of 65 distinct relation types are observed. To better illustrate the structural patterns of GO terms, we visualize the top 10 most frequently occurring relations, as shown in Figure 6.

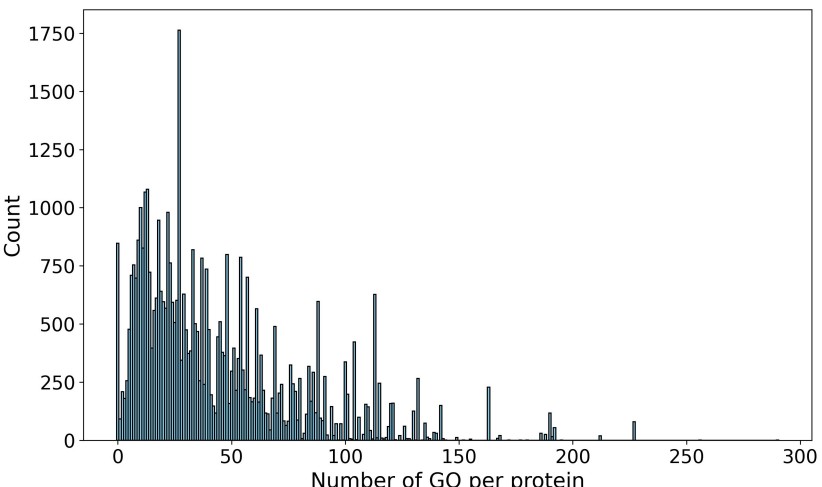

Figure 5: The distribution of the number of unique GO term IDs for each protein. The x-axis shows the counts, and the y-axis shows the number of entries with that count.

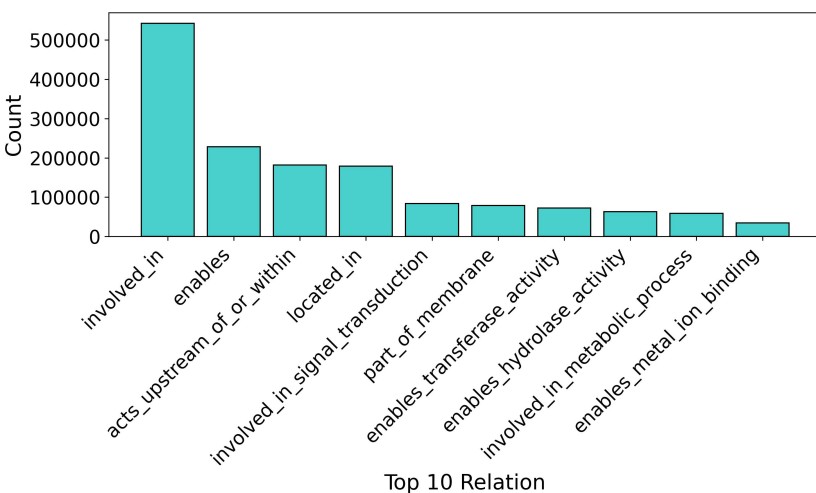

Figure 6: The distribution of the number of relations in P2G triples. Only the top 10 most frequent values are shown.

## C    DETAILS OF VIRTUAL SCREENING BENCHMARKS

**DUD-E benchmark** (Mysinger et al., 2012): The DUD-E (Directory of Useful Decoys, Enhanced) dataset is one of the most commonly used resources for evaluating virtual screening algorithms. It

comprises 102 protein targets and a total of 22,886 active compounds, with an average of approximately 224 actives per target. For each active molecule, 50 decoys are provided, which are designed to share similar physicochemical properties while differing significantly in two-dimensional topology.

**LIT-PCBA benchmark** (Tran-Nguyen et al., 2020): The Large-scale Interaction and Toxicity Prediction for Chemical Bioactivity Assays (LIT-PCBA) dataset offers a more challenging and realistic benchmark compared to DUD-E. It consists of 15 protein targets, 7,844 active compounds, and 407,381 unique inactive molecules. All samples are curated from high-confidence PubChem BioAssay experiments.

**DEKOIS 2.0 benchmark** (Bauer et al., 2013): The DEKOIS 2.0 dataset is designed to provide a stringent evaluation of structure-based virtual screening methods. It comprises 81 protein targets from a variety of protein families. For each target, there are 40 known active ligands and 1,200 decoys that resemble drug-like molecules but are assumed to be inactive.

**AD benchmark** (Chen et al., 2019): The AD benchmark provides an improved evaluation setting by addressing the target-specific decoy bias present in datasets such as DUD-E. Instead of generating synthetic decoys, it uses active compounds from unrelated protein targets as decoys. For each target, active molecules from 101 other targets are screened through docking, and the top 50 based on binding affinity are selected to serve as decoys.

## D   DETAILS OF TRAINING AND INFERENCE

In all experiments that involve structural information, during both training and inference, we use whole-protein structures predicted by AlphaFold2 (Jumper et al., 2021). We intentionally avoid using ligand-bound experimental structures from the Protein Data Bank (Berman et al., 2000) because their conformations are influenced by the co-crystallized ligands. This ligand dependence creates inconsistencies during training, since different ligands can induce different pocket geometries, and it also introduces ligand-specific biases at test time. These biases are incompatible with our intended application scenario, which focuses on whole-protein virtual screening where the binding pocket may be unknown or unreliable. In comparison, AlphaFold2 provides consistent apo-like structural predictions that are free of ligand-induced artifacts and therefore better aligned with the requirements of our setting.

All experiments are conducted on a single NVIDIA A40 GPU. We train each model for 20 epochs using a batch size of 8. The initial learning rate is set to $1 \times 10^{-4}$, with a linear warm-up over the first 6% of total training steps. The optimizer used for all training runs is AdamW.

For the GO-based contrastive loss $\mathcal{L}_{\mathrm{GO}}$, we assign different weights $\epsilon$ to balance its contribution across variants: a weight of 0.1 for Drug-ProGO(seq) and 0.01 for Drug-ProGO(str).

During training, we freeze most layers of the large pre-trained models, ESM2 (Lin et al., 2023) and SaProt (Su et al., 2024), except for the layer normalization layers, which remain trainable. For all other backbones, we perform full fine-tuning of model parameters. This strategy allows us to benefit from the rich representations learned by large-scale protein models while avoiding overfitting in our relatively small downstream dataset.

For inference, the hyperparameter $\lambda$ controlling the sensitivity to uncertainty is set to 1.4.

## E   DETAILS OF TRAINING LOSSES

Given a set of proteins $\mathcal{P} = \{p_i\}$, each $p_i$ is associated with a set of candidate ligands $\mathcal{M}_i = \{m_i^j\}$, which are ranked based on their binding affinity labels. The contrastive loss encourages the model to distinguish active protein–ligand pairs from inactive ones by maximizing the similarity of positive pairs and minimizing it for negative pairs. For each protein $p_i$, the loss includes two components: a pocket-to-ligand loss $\mathcal{L}_{p \to m}^i$ that aligns ligand embeddings toward their corresponding protein embedding, and a ligand-to-pocket loss $\mathcal{L}_{m \to p}^i$ that performs the reverse alignment. Formally, the

Table 7: RE (%) comparison of Drug-ProGO with and without GO across four benchmarks.

| Benchmark | Method | 0.5% | 1% | 2% | 5% |
|---|---|---|---|---|---|
| DUD-E | Drug-ProGO(seq)w/o GO | 113.51 | 62.27 | 34.09 | 15.07 |
| | Drug-ProGO(seq) | 126.17 | 68.48 | 36.64 | 15.78 |
| | Drug-ProGO(str)w/o GO | 120.19 | 65.65 | 35.25 | 15.53 |
| | Drug-ProGO(str) | 128.04 | 69.44 | 37.19 | 16.14 |
| | Drug-ProGO(both)w/o GO | 128.13 | 69.34 | 37.33 | 16.16 |
| | Drug-ProGO(both) | **139.07** | **73.83** | **38.96** | **16.63** |
| LIT-PCBA | Drug-ProGO(seq)w/o GO | 9.88 | 5.64 | 4.52 | 2.96 |
| | Drug-ProGO(seq) | 10.89 | 6.64 | 4.18 | 3.10 |
| | Drug-ProGO(str)w/o GO | 12.14 | 6.94 | 4.61 | 3.44 |
| | Drug-ProGO(str) | 11.64 | 7.39 | 4.43 | 3.18 |
| | Drug-ProGO(both)w/o GO | 10.85 | 6.49 | 5.12 | 3.58 |
| | Drug-ProGO(both) | **12.87** | **7.68** | **4.97** | **3.26** |
| DEKOIS 2.0 | Drug-ProGO(seq)w/o GO | 105.77 | 61.40 | 34.58 | 15.58 |
| | Drug-ProGO(seq) | 124.84 | 69.77 | 38.36 | 16.64 |
| | Drug-ProGO(str)w/o GO | 114.73 | 65.03 | 35.96 | 15.71 |
| | Drug-ProGO(str) | 122.20 | 67.21 | 36.43 | 15.79 |
| | Drug-ProGO(both)w/o GO | 119.42 | 67.93 | 37.66 | 16.41 |
| | Drug-ProGO(both) | **133.05** | **73.16** | **39.11** | **16.86** |
| AD | Drug-ProGO(seq)w/o GO | 71.84 | 46.17 | 28.38 | 14.55 |
| | Drug-ProGO(seq) | 84.95 | 53.68 | 32.17 | 15.39 |
| | Drug-ProGO(str)w/o GO | 79.58 | 50.65 | 20.81 | 15.05 |
| | Drug-ProGO(str) | 82.16 | 52.12 | 31.90 | 15.41 |
| | Drug-ProGO(both)w/o GO | 84.82 | 52.98 | 32.02 | 15.63 |
| | Drug-ProGO(both) | **90.93** | **56.71** | **33.77** | **15.92** |

overall contrastive loss is averaged across all proteins:

$$\mathcal{L}_c = \frac{1}{|\mathcal{P}|} \sum_{i \in \mathcal{P}} \gamma_i \left( \mathcal{L}_{p \to m}^i + \mathcal{L}_{m \to p}^i \right), \tag{6}$$

where $\gamma_i = \frac{1}{\sqrt{|\mathcal{M}_i|}}$ balances the contribution of each protein with $|M_i|$ ligands. For more details, please refer to LigUnity(Feng et al., 2025).

We employ a ranking loss $\mathcal{L}_r$ to enhance the model's ability to correctly order candidate molecules according to their measured binding affinities to a given protein. This loss is derived from the Plackett–Luce model (Cao et al., 2007), which treats ranking as a sequential selection process. For the $i$-th protein target, let $\pi(k)$ denote the ligand ranked at position $k$, and let $M_{\leq k}$ denote the subset of ligands in $M_i$ whose affinities are no higher than that of $\pi(k)$. Accordingly, the ranking loss is defined as the weighted negative log-likelihood of the observed ranking:

$$\mathcal{L}_r^i = - \sum_{k=1}^{|\mathcal{M}_i|} w_k \log \left( \frac{\exp \left( \text{sim}(f_p^i, f_m^{\pi(k)})/\tau \right)}{\sum_{j \in \mathcal{M}_{\leq k}} \exp \left( \text{sim}(f_p^i, f_m^j)/\tau \right)} \right), \tag{7}$$

where $f_p^i$ and $f_m^j$ denote the embeddings of the $i$-th protein and the $j$-th ligand in $M_i$, respectively; $\tau$ is a temperature hyperparameter; $\text{sim}(,)$ denotes a similarity function (e.g., cosine similarity); and $w_k = \frac{1}{\sqrt{|\mathcal{M}_i|}} \frac{1}{\log(k+1)}$ weights higher-ranked ligands more heavily while mitigating the influence of proteins with many candidate ligands.

Lastly, the training loss incorporates a GO-based contrastive objective, which improves protein representations by aligning them with relevant GO terms. Details of the GO enhancement mechanism are described below. We follow the contrastive learning paradigm from LigUnity(Feng et al., 2025) to construct the virtual screening training stage.

## F    EVALUATION METRICS

**BEDROC (Boltzmann-Enhanced Discrimination of Receiver-Operating Characteristic):** BEDROC is a widely used metric in virtual screening, particularly in drug discovery tasks. By introducing an exponential weighting factor, it assigns more importance to early-ranked compounds,

Table 8: Ablation study on the number of GO term sampling and negative triple sampling. Each entry is denoted as goX negY, where X indicates the number of sampled GO terms and Y indicates the number of sampled negative triples. All Drug-ProGO variants are based on go5 neg3. Bold indicates better-performing.

| Method | AUROC (%) | BEDROC (%) | EF 0.5% | EF 1% | EF 5% |
|---|---|---|---|---|---|
| Drug-ProGO(seq)go5 neg3 | **93.51** | **74.68** | **53.71** | **48.08** | **15.49** |
| -go1 neg3 | 92.95 | 71.70 | 51.01 | 45.97 | 15.16 |
| -go3 neg3 | 92.32 | 69.52 | 49.99 | 44.59 | 14.68 |
| -go7 neg3 | 92.46 | 70.01 | 51.42 | 44.78 | 14.66 |
| -go5 neg1 | 93.50 | 72.41 | 52.22 | 46.36 | 15.30 |
| -go5 neg5 | 92.75 | 70.77 | 51.06 | 44.91 | 14.99 |
| -go5 neg7 | 93.13 | 72.22 | 51.48 | 46.03 | 15.42 |
| Drug-ProGO(str)go5 neg3 | **93.75** | **76.21** | **54.42** | **49.31** | **15.80** |
| -go1 neg3 | 93.22 | 71.12 | 49.80 | 45.44 | 15.26 |
| -go3 neg3 | 93.31 | 71.56 | 50.45 | 45.88 | 15.32 |
| -go7 neg3 | 92.96 | 70.32 | 50.50 | 45.07 | 15.07 |
| -go5 neg1 | 93.13 | 72.87 | 52.58 | 47.11 | 15.22 |
| -go5 neg5 | 93.30 | 71.97 | 51.67 | 46.46 | 15.18 |
| -go5 neg7 | 93.37 | 72.31 | 52.02 | 46.53 | 15.39 |
| Drug-ProGO(both)go5 neg3 | **94.61** | **79.79** | **56.51** | **51.84** | **16.33** |
| -go1 neg3 | 94.04 | 76.53 | 53.50 | 49.29 | 15.96 |
| -go3 neg3 | 94.04 | 76.58 | 53.55 | 49.43 | 16.00 |
| -go7 neg3 | 93.96 | 76.14 | 54.68 | 49.19 | 15.73 |
| -go5 neg1 | 94.38 | 77.57 | 55.12 | 50.30 | 16.03 |
| -go5 neg5 | 94.20 | 77.57 | 54.45 | 49.46 | 15.85 |
| -go5 neg7 | 94.41 | 77.86 | 55.45 | 50.09 | 16.26 |

thereby emphasizing the model's ability to identify active molecules early in the screening process. The commonly used variant, $\text{BEDROC}_{85}$, assigns approximately 80% of the score to the top 2% of ranked candidates. The formula for BEDROC is defined as:

$$\text{BEDROC}_\alpha = \frac{\sum\limits_{i=1}^{N} \exp\left(-\frac{\alpha r_i}{N}\right)}{Z_\alpha \left(\frac{1-\exp(-\alpha)}{\exp(\alpha/N)-1}\right)} \times \frac{Z_\alpha \sinh\left(\frac{\alpha}{2}\right)}{\cosh\left(\frac{\alpha}{2}\right) - \cosh\left(\frac{\alpha}{2} - \alpha Z_\alpha\right)} + \frac{1}{1 - \exp\left(\alpha(1 - Z_\alpha)\right)}, \quad (8)$$

where $\alpha$ controls the sensitivity to early-ranked items (typically ranging from 1 to 100), $N$ is the total number of compounds, $r_i$ is the rank of the $i$-th compound, and $Z_\alpha$ is a normalization constant.

**EF (Enrichment Factor):** The Enrichment Factor measures how concentrated the active compounds are among the top-ranked results. A higher EF indicates better model performance in prioritizing active compounds. It is defined as:

$$\text{EF}_\alpha = \frac{\text{NTB}_\alpha}{\alpha \cdot \text{NTB}_t}, \quad (9)$$

where $\text{NTB}_\alpha$ is the number of actives within the top $\alpha$% of the ranked list, and $\text{NTB}_t$ is the total number of active compounds.

**RE (ROC Enrichment):** The ROC Enrichment metric evaluates the ability to enrich true positives while controlling the number of false positives at a specified threshold. It is defined as:

$$\text{RE}(x\%) = \frac{n \cdot \text{TP}}{\text{P} \cdot \text{FP}_{x\%}}, \quad (10)$$

where $n$ is the total number of molecules, TP is the number of true positives, P is the total number of actives, and $\text{FP}_{x\%}$ is the number of false positives at the $x$% cutoff.

To assess the effectiveness of the GO enhancing strategy, we compare Drug-ProGO and Drug-ProGO (w/o GO) across four benchmarks using the RE metric in three input variants ("seq", "str" and "both"). As shown in Table 7, the results demonstrate that incorporating the GO enhancing strategy promotes early enrichment performance.

Table 9: Ablation study on protein encoder training strategies. "all_ft" indicates full fine-tuning of the protein encoder the corresponding model. Bold indicates better-performing.

| Method | AUROC (%) | BEDROC (%) | EF | | |
|---|---|---|---|---|---|
| | | | 0.5% | 1% | 5% |
| Drug-ProGO(seq) all_ft | 92.70 | 69.68 | 49.98 | 44.62 | 14.84 |
| Drug-ProGO(seq) | **93.51** | **74.68** | **53.71** | **48.08** | **15.49** |
| Drug-ProGO(str) all_ft | 93.12 | 70.94 | 50.12 | 45.40 | 15.25 |
| Drug-ProGO(str) | **93.75** | **76.21** | **54.42** | **49.31** | **15.80** |
| Drug-ProGO(both) all_ft | 93.94 | 75.11 | 52.47 | 48.22 | 15.77 |
| Drug-ProGO(both) | **94.61** | **79.79** | **56.51** | **51.84** | **16.33** |

## G ABLATION STUDY

We conduct an ablation study to investigate the impact of different training strategies for the protein encoder. In our method, we freeze all parameters of the protein encoder except for the layer norm weights. To evaluate the effectiveness of this partial fine-tuning strategy, we compare it against full fine-tuning, where all parameters of the protein encoder are updated during training. The results are shown in Table 9.

Furthermore, we conduct an ablation study on the number of sampled GO terms for each protein and the number of sampled negative triples. We evaluate four settings with 1, 3, 5, and 7 entries, respectively. As shown in Table 8 and Figure 7, the setting with 5 GO terms and 3 negative samples achieves the best performance.

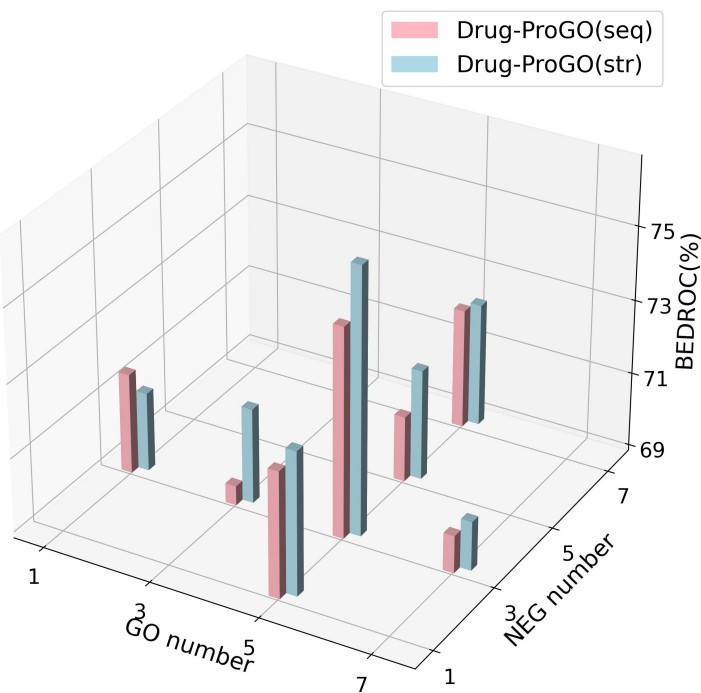

Figure 7: Ablation study on the impact of GO term sampling and negative triple sampling in Drug-ProGO(seq) and Drug-ProGO(str) on the DUD-E benchmark. Each entry is denoted as goX negY, where X indicates the number of sampled GO terms and Y indicates the number of sampled negative triples.

## H ADDITIONAL EXPERIMENTS ON UNSEEN PROTEINS

To further evaluate the generalization ability of our method to unseen proteins, we exclude training proteins that share more than 80% or 40% sequence identity with any test protein on three benchmarks. Protein sequence similarity is calculated using cd-hit (Li & Godzik, 2006), where the similarity is defined as the ratio of identical residues in the aligned sequence. For reference, the case with 100% exclusion corresponds to the standard unseen setting. The results are reported in Tables 10 and 11, corresponding to sequence-based and structure-based Drug-ProGO, respectively.

Table 10: Drug-ProGO(seq)' performance on unseen proteins under different sequence similarity thresholds.

| Method | AUROC (%) | BEDROC (%) | EF 0.5% | EF 1% | EF 5% |
|---|---|---|---|---|---|
| sim40% w/o GO | 76.04 | 36.78 | 27.49 | 23.79 | 8.39 |
| sim40% | **77.99** | **40.25** | **29.49** | **25.72** | **9.12** |
| sim80% w/o GO | 83.37 | 57.14 | 42.15 | 37.54 | 12.08 |
| sim80% | **84.61** | **58.94** | **43.36** | **38.69** | **12.53** |
| sim100% w/o GO | 90.97 | 71.56 | 51.76 | 46.81 | 14.65 |
| sim100% | **91.65** | **75.03** | **53.70** | **49.11** | **15.40** |

Table 11: Drug-ProGO(str)' performance on unseen proteins under different sequence similarity thresholds.

| Method | AUROC (%) | BEDROC (%) | EF 0.5% | EF 1% | EF 5% |
|---|---|---|---|---|---|
| sim40% w/o GO | 80.98 | 40.48 | 29.89 | 25.32 | 9.46 |
| sim40% | **81.14** | **43.40** | **32.02** | **27.63** | **9.93** |
| sim80% w/o GO | 85.83 | 58.22 | 42.36 | 37.91 | 12.48 |
| sim80% | **86.11** | **58.54** | **42.59** | **38.07** | **12.61** |
| sim100% w/o GO | 90.86 | 69.41 | 48.80 | 44.68 | 14.78 |
| sim100% | **91.93** | **71.08** | **50.60** | **45.80** | **14.99** |

From both tables, we observe that the performance decreases as the sequence similarity threshold becomes stricter, which confirms that the task becomes more challenging with less information shared between training and test proteins. Nevertheless, our method consistently outperforms the baseline without GO enhancement across all settings. This demonstrates that GO enhancement provides additional robustness and contributes to better generalization to unseen proteins under diverse similarity constraints.

## I ADDITIONAL EXPERIMENTS ON GO-SIMILARITY

To further strengthen the robustness of our evaluation, we conduct additional experiments that remove training proteins whose GO terms are highly similar to those of any test protein. Concretely, for each DUD-E target, we compare its GO term set with the GO term set of every training protein and compute their functional similarity. Training proteins whose similarity to a test protein exceeded a given threshold are excluded. We examine two thresholds, 80% and 40%. This procedure led to a substantial reduction in training data size: from the original 30,580 assays to 15,279 assays under the 80% threshold and to 7,209 assays under the 40% threshold. The results after filtering are provided in the Table 12.

We note that the overall performance drop is expected because removing GO-similar proteins also eliminates a large portion of the training data. In particular, the 40% threshold reduces the dataset by more than 75%, and this reduction in data volume naturally contributes to the decrease in performance.

Despite this reduction, we consistently observe that adding the GO enhancement improves performance over the corresponding without-GO variants across all settings ("seq", "str", and "both").

This trend holds under both filtering thresholds, indicating that the GO module provides meaningful functional signals rather than relying on any implicit overlap between the training and test proteins.

Under the moderate filtering setting (GO similarity $\geq 80\%$), where the remaining training data are still sufficient for learning, our method continues to outperform the LigUnity-seq baseline by a clear margin. This demonstrates that our approach remains effective even when proteins functionally close to the test set are removed from training.

Table 12: Evaluation of Drug-ProGO under GO-similarity filtering of training proteins at thresholds 80% and 40%.

| Method | AUROC (%) | BEDROC (%) | EF 0.5% | 1% | 5% |
|---|---|---|---|---|---|
| sim40%(seq) w/o go | **79.64** | 37.43 | 25.86 | 23.48 | 9.20 |
| sim40%(seq) | 79.39 | **39.21** | **26.54** | **24.38** | **9.51** |
| sim40%(str) w/o go | **80.49** | 37.61 | 25.72 | 23.17 | 9.09 |
| sim40%(str) | 80.19 | **41.76** | **28.51** | **25.72** | **10.00** |
| sim40%(both) w/o go | **83.26** | 43.28 | 29.19 | 26.60 | 10.44 |
| sim40%(both) | 82.38 | **46.04** | **31.24** | **28.54** | **10.79** |
| sim80%(seq) w/o go | 87.11 | 62.30 | 44.02 | 40.15 | 13.35 |
| sim80%(seq) | **88.82** | **65.12** | **46.40** | **42.30** | **13.75** |
| sim80%(str) w/o go | 89.70 | 65.98 | 47.02 | 42.72 | 14.16 |
| sim80%(str) | **90.33** | **68.07** | **48.88** | **44.26** | **14.39** |
| sim80%(both) w/o go | 90.32 | 68.36 | 47.38 | 44.15 | 14.58 |
| sim80%(both) | **91.36** | **70.91** | **49.92** | **46.01** | **14.90** |

## J   PERFORMANCE COMPARISON OF DIFFERENT INPUTS

To better understand the contribution of different input modalities, we analyzed the performance of our model under sequence-only, structure-only and both settings across all DUD-E targets. As shown in Figure 8, although the combined input generally yields the best performance on average, sequence-only input still provides meaningful predictions in certain cases.

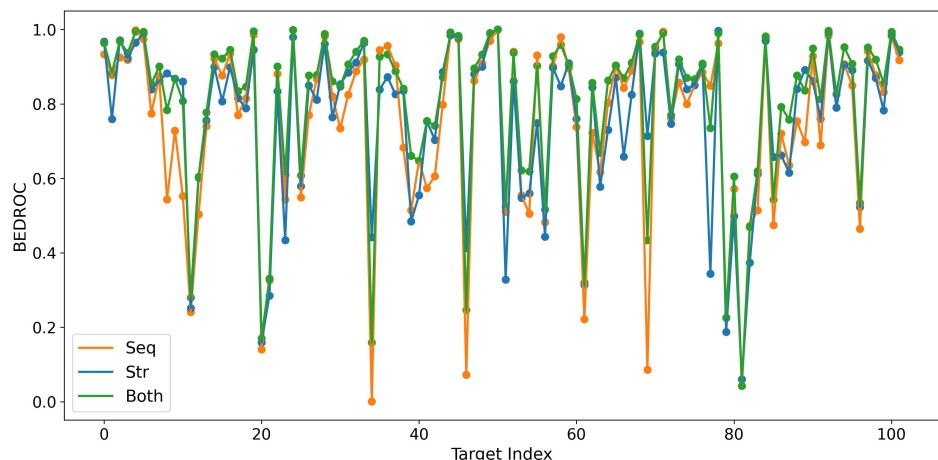

Figure 8:   BEDROC values for each DUD-E target under different input settings: sequence-only (Seq), structure-only (Str), and combined (Both).

## K CASE STUDY

As shown in Figure 9, CP3A4 (Werk & Cascorbi, 2014) is a representative unseen target in our test set, characterized by an unusually large and highly plastic binding region with significant conformational flexibility. Unlike typical targets with well-defined deep binding pockets, CP3A4 lacks a distinct cavity; instead, small molecules tend to "float" within an expansive hydrophobic channel. This inherent pocket plasticity presents a major challenge for conventional pocket-based virtual screening approaches. Our method, Drug-ProGO, surpasses the SOTA pocket-input model LigUnity on this target, as traditional pocket definitions are often inaccurate or incomplete for CP3A4. Importantly, integrating GO information markedly enhances the performance of both sequence- and structure-based versions of our model. Specifically, we observe an approximate +41.41% improvement in BEDROC scores when comparing Drug-ProGO(seq) against the variant without GO, underscoring the efficacy of GO-based functional supervision in addressing proteins with ambiguous or flexible binding sites. These results further show the strong generalization ability of Drug-ProGO to unseen proteins.

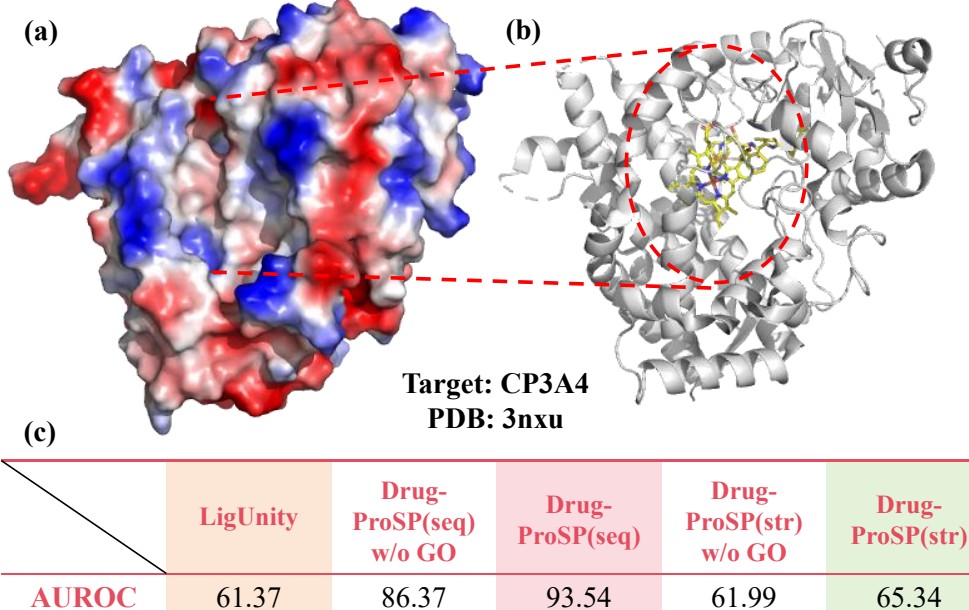

|  | LigUnity | Drug-ProSP(seq) w/o GO | Drug-ProSP(seq) | Drug-ProSP(str) w/o GO | Drug-ProSP(str) |
|---|---|---|---|---|---|
| **AUROC** | 61.37 | 86.37 | 93.54 | 61.99 | 65.34 |
| **BEDROC** | 18.54 | 35.59 | 77.00 | 26.87 | 28.51 |

Figure 9: (a) Structural representation of CP3A4 (PDB ID: 3nxu) from the DUD-E benchmark, highlighting its large and flexible binding region. (b) Known small molecules tend to "float" within a broad hydrophobic channel. (c) Comparative screening performance of multiple methods on CP3A4, measured by AUROC and BEDROC metrics, demonstrating the superior performance of Drug-ProGO.

## L LLM USAGE STATEMENT

In this work, LLMs are only employed as writing assistants for language polishing and minor editing. They are not involved in research ideation, experimental design, analysis, or result generation.

