# OpenReview forum: "Drug-ProGO: A Gene Ontology-Enhanced Contrastive Learning Framework for Drug Virtual Screening with Multi-modality Whole-Protein Input"
_ICLR.cc/2026/Conference — Submitted to ICLR 2026_

### Official Review · Reviewer_pYPJ · 2025-10-29

**Soundness:** 3
**Presentation:** 3
**Contribution:** 2
**Rating:** 4
**Confidence:** 5

**Summary:**

This paper introduces a new method for dense retrieval basd virtual screening called Drug-ProGO. Building upon prior work including DrugCLIP and LigUnity, this work introduce a new Gene Ontology (GO) enhancement module designed to enrich protein representations by incorporating functional biological knowledge. This module is integrated into a contrastive learning framework to improve the alignment between protein and ligand embeddings. The overall training strategy, datasets, and encoder backbones largely follow those used in previous studies, but the addition of GO-based supervision aims to boost generalization to unseen proteins. Empirical results show that Drug-ProGO achieves competitive or improved performance compared to baselines across multiple virtual screening benchmarks.

**Strengths:**

1. The use of Gene Ontology information to enhance protein representations for virtual screening is a reasonable and meaningful idea, especially for improving generalization to unseen proteins.
2. The figures in the paper are well-designed and visually clear, helping to convey the framework and key concepts effectively.
3.  Ablation studies demonstrate that incorporating GO information does lead to measurable improvements in performance.

**Weaknesses:**

1. The improvements of Drug-ProGO over previous methods such as LigUnity are quite limited, especially on the three datasets other than DUD-E. Even in the sequence only setting, the performance gains of Drug-ProGO compared to LigUnity are marginal.
2. Drug-ProGO is primarily an incremental work built upon existing approaches. The main novelty lies in the integration of Gene Ontology information. While this idea is reasonable and does lead to some improvements, the overall technical novelty and contribution of the work remain limited.
3. There should be a more thorough analysis of how GO contributes to the performance improvement. For example, the visualization in Figure 3 shows better embeddings for some targets, but it does not clearly illustrate how these improvements are connected to the GO information.
4. The comparison on unseen proteins in Table 5 is a good addition, but further analysis would be valuable, such as stratifying the evaluation based on sequence identity or pocket similarity to better understand generalization across proteins with varying degrees of similarity.

**Questions:**

see Weaknesses

---

> ### Author Response · Authors · 2025-11-20
> **Response to W1 and W2**
>
> Thank you for taking the time to review our paper and for your thoughtful and thorough feedback. We truly appreciate your recognition that our idea is reasonable and meaningful. Below, we address your questions and concerns in detail.
>
> ---
>
> > **W1. The improvements of Drug-ProGO over previous methods such as LigUnity are quite limited, especially on the three datasets other than DUD-E. Even in the sequence only setting, the performance gains of Drug-ProGO compared to LigUnity are marginal.**
>
> Thank you for your observation. We acknowledge that the performance improvement of Drug-ProGO over LigUnity is relatively modest on datasets other than DUD-E. This is primarily because LigUnity uses pocket-level inputs, which provide more explicit information, whereas our method operates on whole-protein inputs, which are inherently more challenging. Despite this, Drug-ProGO consistently demonstrates improvements across most evaluation metrics, showing that it captures additional signal beyond what LigUnity can leverage.
>
> More importantly, one of the key contributions of our work is that it substantially broadens the scope of virtual screening. Drug-ProGO can be applied to proteins for which only the sequence is available or proteins without known pocket annotations. LigUnity and other pocket-based methods cannot operate in such settings. By enabling screening for these targets, our framework addresses a much wider and more realistic range of proteins commonly encountered in early-stage drug discovery.
>
> We revise our paper to make our contribution crystal clear. Thank you again for your insightful comment.
>
> ---
>
> > **W2. Drug-ProGO is primarily an incremental work built upon existing approaches. The main novelty lies in the integration of Gene Ontology information. While this idea is reasonable and does lead to some improvements, the overall technical novelty and contribution of the work remain limited.**
>
> Thank you for raising this important point. We agree that part of the performance improvement comes from incorporating GO information. However, the role of GO in our framework is not a naive feature augmentation. GO provides function-level supervision that complements sequence and structure features, enabling the model to capture biological relationships that are invisible to purely geometric similarity or evolutionary homology.
>
> At the same time, **our GO integration mechanism provides a meaningful and non-trivial technical contribution**. Since GO annotations are typically unavailable during inference, the model must benefit from GO supervision only during training while relying solely on sequence or structure inputs at test time. This design requires aligning protein representations with functional semantics rather than directly concatenating GO vectors. Moreover, leveraging GO in a meaningful way requires integrating both protein-to-GO and GO-to-GO relations while maintaining the modality consistency required by the CLIP framework. These considerations make GO integration substantially more complex than simply adding another information source.
>
> Moreover, **the performance gains from GO largely stem from its positive effect on generalization toward unseen proteins**. When sequence or structural similarity between training and test proteins is low, GO acts as a functional bridge that brings unseen proteins closer to relevant regions of the representation space. Our embedding-distance analysis confirms that models trained with GO consistently reduce the distance between unseen test proteins and their nearest training neighbors, indicating that GO reshapes the representation geometry rather than providing a superficial boost.
>
> Finally, we emphasize that GO is only one part of our contribution; the primary novelty of Drug-ProGO lies in substantially expanding the applicability of virtual screening beyond the pocket-based setting. **Our framework supports sequence-only screening, structure-only screening, and joint sequence–structure screening**, thereby enabling virtual screening for a broader and more realistic range of protein targets, including those without pocket annotations or reliable structures. This expanded applicability represents a major conceptual and practical advance over existing pocket-specific approaches.
>
> We revise the manuscript to clarify these points, and the updated content is highlighted in blue.
>
> ---
>
> > **Due to space limitations, our responses to W3 and W4 are provided in the next box.**

---

> ### Author Response · Authors · 2025-11-20
> **Response to W3 and W4**
>
> > **W3. There should be a more thorough analysis of how GO contributes to the performance improvement. For example, the visualization in Figure 3 shows better embeddings for some targets, but it does not clearly illustrate how these improvements are connected to the GO information.**
>
> To further analyze how GO improves model performance, we visualize the distance between each DUD-E target and its nearest training protein embedding, comparing models trained with and without GO enhancement for both sequence and structure inputs. As shown in the figure (https://pasteboard.co/z8FQgL3WZPTQ.jpg), introducing GO consistently reduces these distances, indicating that unseen test proteins become closer to the training distribution. This suggests that GO acts as a functional bridge that reshapes the representation space, allowing proteins with low sequence or structural similarity to align with biologically relevant neighbors and thereby improving generalization in virtual screening.
>
> We add this analysis to Section 3.4 of the revised manuscript, highlighted in blue.
>
> ---
>
> > **W4. The comparison on unseen proteins in Table 5 is a good addition, but further analysis would be valuable, such as stratifying the evaluation based on sequence identity or pocket similarity to better understand generalization across proteins with varying degrees of similarity.**
>
> Thank you for your comment. In the original manuscript, we provide a detailed analysis of sequence similarity in Appendix H. For convenience, we summarize here that Drug-ProGO’s performance on unseen proteins remains robust across varying levels of sequence identity and pocket similarity, indicating strong generalization ability. Stratifying evaluation by these factors further confirms that the functional supervision provided by GO is particularly beneficial when test proteins have low similarity to the training set. For convenience, we copy the relevant results below.
>
>
> > Performance of Drug-ProGO(seq) under Different Sequence Similarity Thresholds
> | Method         | AUROC (%) | BEDROC (%) | EF 0.5%   | EF 1%     | EF 5%     |
> | -------------- | --------- | ---------- | --------- | --------- | --------- |
> | sim40% w/o GO  | 76.04     | 36.78      | 27.49     | 23.79     | 8.39      |
> | sim40%     | **77.99** | **40.25**  | **29.49** | **25.72** | **9.12**  |
> | sim80% w/o GO  | 83.37     | 57.14      | 42.15     | 37.54     | 12.08     |
> | sim80%     | **84.61** | **58.94**  | **43.36** | **38.69** | **12.53** |
> | sim100% w/o GO | 90.97     | 71.56      | 51.76     | 46.81     | 14.65     |
> | sim100%    | **91.65** | **75.03**  | **53.70** | **49.11** | **15.40** |
>
> > Performance of Drug-ProGO(str) under Different Sequence Similarity Thresholds**
> | Method         | AUROC (%) | BEDROC (%) | EF 0.5%   | EF 1%     | EF 5%     |
> | -------------- | --------- | ---------- | --------- | --------- | --------- |
> | sim40% w/o GO  | 80.98     | 40.48      | 29.89     | 25.32     | 9.46      |
> |sim40%    | **81.14** | **43.40**  | **32.02** | **27.63** | **9.93**  |
> | sim80% w/o GO  | 85.83     | 58.22      | 42.36     | 37.91     | 12.48     |
> | sim80%     | **86.11** | **58.54**  | **42.59** | **38.07** | **12.61** |
> | sim100% w/o GO | 90.86     | 69.41      | 48.80     | 44.68     | 14.78     |
> | sim100%    | **91.93** | **71.08**  | **50.60** | **45.80** | **14.99** |
>
> ---
>
> From both tables, we observe that the performance decreases as the sequence similarity threshold becomes stricter, confirming that the task becomes more challenging with less information shared between training and test proteins. Nevertheless, our method consistently outperforms the baseline without GO enhancement across all settings. This demonstrates that GO enhancement provides additional robustness and contributes to better generalization to unseen proteins under diverse similarity constraints.

---

### Official Review · Reviewer_oF9z · 2025-10-31

**Soundness:** 3
**Presentation:** 3
**Contribution:** 3
**Rating:** 4
**Confidence:** 4

**Summary:**

This paper proposes a method that integrates Gene Ontology (GO) knowledge into contrastive learning for drug screening without using binding pocket data. The framework combines sequence and structure protein encoders with a molecular encoder and a GO-based contrastive module that links proteins and GO terms. It uses multi-modal inputs, contrastive and ranking losses, and an uncertainty-aware fusion of sequence and structure predictions to improve protein–ligand matching and generalization.

**Strengths:**

The paper is clearly written, and incorporating GO information is a novel attempt. It introduces additional information, and the experiments are relatively thorough.

**Weaknesses:**

The main issue is that the contribution of this paper is relatively incremental, as it mainly adds a new source of information on top of the previous LigUnity framework to align protein representations. How the GO data improves the model’s performance in virtual screening, and what specific information it introduces, deserve further analysis and discussion.

**Questions:**

1.Regarding the score fusion, have you explored other fusion mechanisms? For example, introducing some additional learnable parameters.

2. Would opening the weights of the pre-trained model for full fine-tuning lead to better results?

3.The use of SaProt in this paper differs from structure encoder in LigUnity. Compared with pocket-based approaches, what advantages does it offer?

---

> ### Author Response · Authors · 2025-11-20
> **Response to Weaknesses**
>
> Thank you very much for your careful review and for your positive assessment of our writing quality and our attempt to introduce GO information. We respond to each point in detail below.
>
> > **W1.1 The main issue is that the contribution of this paper is relatively incremental, as it mainly adds a new source of information on top of the previous LigUnity framework to align protein representations.**
>
> Thank you for raising this important point. We agree that part of the performance improvement comes from incorporating GO information. However, the role of GO in our framework is not a naive feature augmentation. GO provides function-level supervision that complements sequence and structure features, enabling the model to capture biological relationships that are invisible to purely geometric similarity or evolutionary homology.
>
> At the same time, **our GO integration mechanism provides a meaningful and non-trivial technical contribution**. Since GO annotations are typically unavailable during inference, the model must benefit from GO supervision only during training while relying solely on sequence or structure inputs at test time. This design requires aligning protein representations with functional semantics rather than directly concatenating GO vectors. Moreover, leveraging GO in a meaningful way requires integrating both protein-to-GO and GO-to-GO relations while maintaining the modality consistency required by the CLIP framework. These considerations make GO integration substantially more complex than simply adding another information source.
>
> Moreover, **the performance gains from GO largely stem from its positive effect on generalization toward unseen proteins**. When sequence or structural similarity between training and test proteins is low, GO acts as a functional bridge that brings unseen proteins closer to relevant regions of the representation space. Our embedding-distance analysis confirms that models trained with GO consistently reduce the distance between unseen test proteins and their nearest training neighbors, indicating that GO reshapes the representation geometry rather than providing a superficial boost.
>
> Finally, we emphasize that GO is only one part of our contribution; the primary novelty of Drug-ProGO lies in substantially expanding the applicability of virtual screening beyond the pocket-based setting. **Our framework supports sequence-only screening, structure-only screening, and joint sequence-structure screening**, thereby enabling virtual screening for a broader and more realistic range of protein targets, including those without pocket annotations or reliable structures. This expanded applicability represents a major conceptual and practical advance over existing pocket-specific approaches.
>
> We revise the manuscript to clarify these points, and the updated content is highlighted in blue.
>
> > **W1.2 How the GO data improves the model’s performance in virtual screening, and what specific information it introduces, deserve further analysis and discussion.**
>
> To further analyze how GO improves model performance, we visualize the distance between each DUD-E target and its nearest training protein embedding, comparing models trained with and without GO enhancement for both sequence and structure inputs. As shown in the figure (https://pasteboard.co/z8FQgL3WZPTQ.jpg), introducing GO consistently reduces these distances, indicating that unseen test proteins become closer to the training distribution. This suggests that GO acts as a functional bridge that reshapes the representation space, allowing proteins with low sequence or structural similarity to align with biologically relevant neighbors and thereby improving generalization in virtual screening.
>
> We add this analysis to Section 3.4 of the revised manuscript, highlighted in blue.
>
> ---
> > **Due to space limitations, our responses to the questions are provided in the next box.**

---

> ### Author Response · Authors · 2025-11-20
> **Response to Questions**
>
> > **Q1. Regarding the score fusion, have you explored other fusion mechanisms? For example, introducing some additional learnable parameters.**
>
> Thank you for the suggestion. Following your advice, we experiment with a learnable fusion mechanism by freezing both the sequence and structure encoders and adding a small trainable MLP head to combine their scores. The fusion head is trained using a ranking loss.The results are as follows:
>
> | Method            | AUCROC (%) | BEDROC (%) | EF 0.50% | EF 1% | EF 5% |
> |-------------------|------------|-------------|----------|--------|--------|
> | Learned head       | 93.31     | 74.14       | 53.26   | 47.74 | 15.43 |
> | Drug-ProGO (both)  | **94.61** | **79.79**   | **56.51** | **51.84** | **16.33** |
>
> The learnable fusion does not outperform our parameter-free fusion. We believe this occurs partly because the structure and sequence encoders are trained under different upstream objectives and therefore introduce mild distribution mismatch; a trainable fusion head may overfit to these inconsistencies. The observed results thus support our choice of a lightweight, non-parametric score-fusion design. We add this ablation study to Section 3.3 of the revised manuscript, highlighted in blue.
>
> ---
> > **Q2. Would opening the weights of the pre-trained model for full fine-tuning lead to better results?**
>
> Thank you for raising this question. We conduct a full fine-tuning ablation study in Appendix G of the original manuscript. For convenience, we copy the relevant results below in this response.
>
> > We conduct an ablation study to investigate the impact of different training strategies for the protein encoder. In our method, we freeze all parameters of the protein encoder except for the layer norm weights. To evaluate the effectiveness of this partial fine-tuning strategy, we compare it against full fine-tuning, where all parameters of the protein encoder are updated during training. The results are summarized in the table below:
> | Method                     | AUROC (%) | BEDROC (%) | EF 0.5% | EF 1% | EF 5% |
> |-----------------------------|-----------|------------|---------|-------|-------|
> | Drug-ProGO(seq) all_ft      | 92.70     | 69.68      | 49.98   | 44.62 | 14.84 |
> | Drug-ProGO(seq)        | **93.51** | **74.68**  | **53.71** | **48.08** | **15.49** |
> | Drug-ProGO(str) all_ft      | 93.12     | 70.94      | 50.12   | 45.40 | 15.25 |
> | Drug-ProGO(str)        | **93.75** | **76.21**  | **54.42** | **49.31** | **15.80** |
> | Drug-ProGO(both) all_ft     | 93.94     | 75.11      | 52.47   | 48.22 | 15.77 |
> | Drug-ProGO(both)        | **94.61** | **79.79**  | **56.51** | **51.84** | **16.33** |
>
> From these results, we observe that full fine-tuning does not outperform our partial fine-tuning strategy.
>
> ---
>
> > **Q3. The use of SaProt in this paper differs from structure encoder in LigUnity. Compared with pocket-based approaches, what advantages does it offer?**
>
> Thank you for the insightful question. The structural encoder used in the referenced framework is a pocket-level Uni-Mol encoder, which requires a well-defined binding pocket and operates at the atomic level. This design makes it unsuitable for the whole-protein scenario we target, where binding pockets are unknown or unreliable. Our use of SaProt is motivated precisely by this setting. SaProt is designed for global protein-structure encoding and therefore enables virtual screening even for proteins without pocket annotations. In this sense, the use of SaProt is not only technically appropriate but also essential for extending the applicability of our framework.
>
> ---

---

### Official Review · Reviewer_a8ga · 2025-10-31

**Soundness:** 3
**Presentation:** 3
**Contribution:** 2
**Rating:** 6
**Confidence:** 3

**Summary:**

This paper presents Drug-ProGO, a novel virtual screening framework that enhances protein representations using Gene Ontology (GO) information within a contrastive learning framework. The method supports multi-modality protein inputs (sequence, structure, or both) and demonstrates sota performance across multiple benchmarks.

**Strengths:**

1. The incorporation of Gene Ontology (GO) terms into the contrastive learning framework is a novel approach that effectively enriches protein representations. This integration allows the model to capture functional similarities between proteins, which is crucial for generalizing to unseen proteins and improving virtual screening performance.

2. Drug-ProGO supports flexible protein inputs, including sequence, structure, or both. This flexibility is particularly valuable in scenarios where only sequence information is available or when binding pocket structures are unknown.

3. The method achieves state-of-the-art results on multiple virtual screening benchmarks, including DUD-E, LIT-PCBA, DEKOIS 2.0, and AD.

**Weaknesses:**

1. The method relies heavily on pre-trained models (e.g., ESM2 for sequence encoding and SaProt for structure encoding).  The freezing of most layers during fine-tuning might limit the model's ability to fully adapt to the specific virtual screening task.

2. The method also mentions that contrastive learning can fully utilize paired small molecule and protein data without the need for affinity labels. However, the ranking component of the method itself leverages these labels. This might suggest to show that for a large dataset, if using a ranking loss for the labeled portion while not using it for the unlabeled portion could potentially perform better than not using the ranking loss at all.

3. The method emphasizes that the sequence can be used as input when protein structures are unavailable. However, I think protein structures are not really an issue (thanks to AlphaFold). So, I wonder what would happen to the performance if we used the protein structures decoded by AlphaFold as input instead?

**Questions:**

Refer to the weakness part.

**Details Of Ethics Concerns:**

NA.

---

> ### Author Response · Authors · 2025-11-20
> **Response to Weaknesses**
>
> Thank you for taking the time to review our paper and for your thoughtful feedback. We greatly appreciate your recognition of our novel GO integration, flexible protein inputs, and state-of-the-art performance. Below, we provide detailed responses to your concerns.
>
> ---
>
> >   **W1: Ablation study on fully fine-tuning**
>
> Thank you for raising this question. We conduct a full fine-tuning ablation study in Appendix G of the original manuscript. For convenience, we copy the relevant results below in this response.
>
> > We conduct an ablation study to investigate the impact of different training strategies for the protein encoder. In our method, we freeze all parameters of the protein encoder except for the layer norm weights. To evaluate the effectiveness of this partial fine-tuning strategy, we compare it against full fine-tuning, where all parameters of the protein encoder are updated during training. The results are summarized in the table below:
> | Method                     | AUROC (%) | BEDROC (%) | EF 0.5% | EF 1% | EF 5% |
> |-----------------------------|-----------|------------|---------|-------|-------|
> | Drug-ProGO(seq) all_ft      | 92.70     | 69.68      | 49.98   | 44.62 | 14.84 |
> | Drug-ProGO(seq)        | **93.51** | **74.68**  | **53.71** | **48.08** | **15.49** |
> | Drug-ProGO(str) all_ft      | 93.12     | 70.94      | 50.12   | 45.40 | 15.25 |
> | Drug-ProGO(str)        | **93.75** | **76.21**  | **54.42** | **49.31** | **15.80** |
> | Drug-ProGO(both) all_ft     | 93.94     | 75.11      | 52.47   | 48.22 | 15.77 |
> | Drug-ProGO(both)        | **94.61** | **79.79**  | **56.51** | **51.84** | **16.33** |
>
> From these results, we observe that full fine-tuning does not outperform our partial fine-tuning strategy.
>
> ---
>
> > **W2: Ranking loss and contrastive learning strategy**
>
> Thank you for the insightful comment. We apologize that our original description may not have been sufficiently clear. In fact, our training setup already follows exactly the strategy you suggested: we use the ranking loss on data for which valid affinity order can be established, while relying solely on contrastive learning for all remaining pairs.
>
> Specifically, as in LigUnity, interactions from ChEMBL and BindingDB entries that share the same assay are supervised with the ranking loss, whereas PDBBind samples without assay-consistent affinity labels are treated as unlabeled and trained purely through contrastive objectives. Therefore, our current implementation indeed uses ranking supervision on the labeled subset and contrastive learning on the unlabeled portion, fully aligning with the your suggestion.
>
> We clarify this point in the revised manuscript, highlighted in blue.
>
> ---
>
> > **W3: Use of alphaFold structures and combined inputs**
>
> Thank you for raising this question. We apologize for not making this sufficiently clear in the original submission. In all experiments that utilize structural information, including both training and inference, we use whole-protein structures predicted by AlphaFold2[1]. We intentionally avoid using ligand-bound experimental structures from the Protein Data Bank[2] because their conformations are influenced by co-crystallized ligands. This ligand dependence creates inconsistencies during training, as different ligands induce different pocket geometries, and it also introduces ligand-specific biases at test time. These biases are incompatible with our intended application scenario, which focuses on whole-protein virtual screening where the binding pocket may be unknown or unreliable. In comparison, AlphaFold2 provides consistent apo-like structural predictions that are free of ligand-induced artifacts and therefore better aligned with the requirements of our setting.
>
> Our experiments show that combining sequence and AlphaFold structures yields the best overall performance. This suggests that, in practical applications where both the sequence and a predicted structure are available, using them together is the recommended choice because the two modalities provide complementary biological information. Sequence captures evolutionary and functional patterns, while structure provides geometric constraints that improve binding-related discrimination.
>
> Although the combined setting performs best on average, sequence-only input remains meaningful in practical applications. As illustrated in the figure (https://pasteboard.co/bXaCfZuDUKMw.png), while most DUD-E targets benefit more from the combined input, there are still cases where sequence-only input outperforms, highlighting its continued relevance.
>
> We clarify these points in the revised manuscript, with the corresponding updates marked in blue.
>
> **References**
>
> [1] Jumper J, Evans R, Pritzel A, et al. Highly accurate protein structure prediction with AlphaFold[J]. nature, 2021, 596(7873): 583-589.
>
> [2] Berman H M, Westbrook J, Feng Z, et al. The protein data bank[J]. Nucleic acids research, 2000, 28(1): 235-242.

---

> > ### Comment · Reviewer_a8ga · 2025-11-25
> >
> > Thanks for the reply and I will keep my positive score.

---

> > > ### Author Response · Authors · 2025-11-27
> > >
> > > We sincerely appreciate your prompt and positive feedback.

---

### Official Review · Reviewer_X8Se · 2025-11-04

**Soundness:** 3
**Presentation:** 3
**Contribution:** 3
**Rating:** 6
**Confidence:** 5

**Summary:**

The paper presents Drug-ProGO, a protein-level virtual screening framework that integrates Gene Ontology information into a contrastive learning setup to improve generalization to unseen proteins. By leveraging functional annotations during training, the model learns representations that capture functional similarity among proteins, enabling more accurate prediction of compatible small molecules. The framework supports flexible protein inputs, including sequence, structure, and their combination, and employs an uncertainty-aware fusion mechanism to combine predictions from different modalities. Experiments on four virtual screening benchmarks demonstrate that incorporating GO knowledge consistently enhances performance, particularly for novel proteins lacking binding pocket information.

**Strengths:**

- Effective use of GO information to provide functional supervision and improve model generalization.
- Flexible design supporting multiple input modalities (sequence, structure, or both).
- Uncertainty-aware fusion mechanism offers a principled way to combine predictions without additional training.
- Strong experimental validation across multiple benchmarks with consistent gains in generalization.

**Weaknesses:**

- Figure 2 is not rendered correctly.
- In the experiments, the authors remove protein sequences identical to those in the training set and sequences with high similarity. It would further strengthen the robustness of the evaluation to also exclude proteins that are close in the GO graph.
- Minor: The double helices in Figure 2 resemble DNA strands rather than protein structures.

**Questions:**

- In Figure 3, it appears that GO information primarily benefits sequence-only cases. Would incorporating AlphaFold-predicted structures provide additional improvements compared to GO features?
- Can GO information also be utilized during inference? This could offer a significant advantage for real-world virtual screening, where drug targets are typically well-defined within human biological pathways.

---

> ### Author Response · Authors · 2025-11-20
> **Response to Weaknesses**
>
> Thank you for taking the time to review our paper and for your thoughtful feedback. We greatly appreciate your recognition of our  effective use of GO information and the strong experimental validation. Below, we provide detailed responses to your concerns.
>
> ---
> >  **W1 & W3 Figure rendering correction**
>
> Thank you for the careful inspection. We correct the rendering issues in Figure 2 and replace the double-helix icons with more appropriate graphical elements representing protein structures in the revised manuscript. For your convenience, we provide a link to the updated figure: (https://pasteboard.co/DcyBXCSUthAF.jpg).
>
> ---
> >  **W2 Evaluation by filtering GO-similar targets**
>
> Thank you for your valuable suggestion. To further strengthen the robustness of our evaluation, we conduct additional experiments that remove training proteins whose GO terms are highly similar to those of any test protein. Specifically, for each DUD-E target, we compare its GO term set with the GO term set of every training protein and compute their similarity. Training proteins whose similarity to a test protein exceeds a given threshold are excluded. We examine two thresholds, 80% and 40%. This procedure reduces the training data size substantially: from the original 30,580 assays to 15,279 assays under the 80% threshold and to 7,209 assays under the 40% threshold.
>
> | Method             | AUCROC (%) | BEDROC (%) | EF 0.50% | EF 1% | EF 5% |
> |-------------------|------------|------------|-----------|-------|-------|
> | sim40%(seq) w/o GO  | **79.64**      | 37.43      | 25.86     | 23.48 | 9.20  |
> | sim40%(seq)         | 79.39      | **39.21**      | **26.54**     | **24.38** | **9.51**  |
> | sim40%(str) w/o GO  | **80.49**      | 37.61      | 25.72     | 23.17 | 9.09  |
> | sim40%(str)         | 80.19      | **41.76**      | **28.51**     | **25.72** | **10.00** |
> | sim40%(both) w/o GO | **83.26**      | 43.28      | 29.19     | 26.60 | 10.44 |
> | sim40%(both)        | 82.38      | **46.04**      | **31.24**     | **28.54** | **10.79** |
> | sim80%(seq) w/o GO  | 87.11      | 62.30      | 44.02     | 40.15 | 13.35 |
> | sim80%(seq)         | **88.82**     | **65.12**      | **46.40**     | **42.30** | **13.75** |
> | sim80%(str) w/o GO  | 89.70      | 65.98      | 47.02     | 42.72 | 14.16 |
> | sim80%(str)         | **90.33**      | **68.07**      | **48.88**     | **44.26** | **14.39** |
> | sim80%(both) w/o GO | 90.32      | 68.36      | 47.38     | 44.15 | 14.58 |
> | sim80%(both)        | **91.36**      | **70.91**      | **49.92**     | **46.01** | **14.90** |
>
> Despite this reduction, we consistently observe that adding the GO enhancement improves performance over the corresponding without-GO variants across all settings (“seq”, “str”, and “both”). This trend holds under both filtering thresholds, indicating that the GO module provides meaningful functional signals rather than relying on any implicit overlap between the training and test proteins. Under the moderate filtering setting (GO similarity ≥ 80%), where the remaining training data are sufficient for learning, our method continues to outperform the LigUnity-seq baseline by a clear margin. This demonstrates that our approach remains effective even when proteins functionally close to the test set are removed from training.
>
> Finally, we emphasize that in real-world virtual screening applications, test proteins often share functional proximity with proteins used for training. Importantly, our method does not require any GO information at inference time. Therefore, even if a test protein happens to be functionally related to training proteins, this does not raise any data leakage concerns and reflects the typical practical usage of virtual screening systems.
>
> We incorporate these clarifications and the new experimental results into Appendix I of the revised manuscript.
>
> ---
> > **Due to space limitations, our responses to the questions are provided in the next box.**

---

> > ### Author Response · Authors · 2025-11-20
> > **Response to Questions**
> >
> > >  **Q1. In Figure 3, it appears that GO information primarily benefits sequence-only cases. Would incorporating AlphaFold-predicted structures provide additional improvements compared to GO features?**
> >
> > Thank you for raising this question. We acknowledge that the improvement brought by GO appears more prominent in the sequence-only setting in Figure 3. This observation is expected rather than a limitation. The sequence encoder naturally provides weaker representations compared to the structure encoder, so GO contributes functional information that compensates for the limited expressive power of sequence-only inputs. In contrast, structure-based models already start from a stronger baseline, and the relative gain from GO therefore appears smaller, although the improvement remains consistent.
> >
> > Regarding the use of protein structures, we apologize for not making this sufficiently clear in the original submission. In all experiments that utilize structural information, including both training and inference, we use whole-protein structures predicted by AlphaFold2 [1]. We intentionally avoid using ligand-bound experimental structures from the Protein Data Bank [2] because their conformations are influenced by the co-crystallized ligands. This ligand dependence creates inconsistencies during training, since different ligands can induce different pocket geometries, and it also introduces ligand-specific biases at test time. These biases are incompatible with our intended application scenario, which focuses on whole-protein virtual screening where the binding pocket may be unknown or unreliable. In comparison, AlphaFold2 provides consistent apo-like structural predictions that are free of ligand-induced artifacts and therefore better aligned with the requirements of our setting.
> >
> > We clarify these points in the Appendix D of revised manuscript highlighting in blue.
> >
> > **References**
> >
> > [1] Jumper J, Evans R, Pritzel A, et al. Highly accurate protein structure prediction with AlphaFold[J]. nature, 2021, 596(7873): 583-589.
> >
> > [2] Berman H M, Westbrook J, Feng Z, et al. The protein data bank[J]. Nucleic acids research, 2000, 28(1): 235-242.
> >
> > >  **Q2. Can GO information also be utilized during inference? This could offer a significant advantage for real-world virtual screening, where drug targets are typically well-defined within human biological pathways.**
> >
> > Thank you for your insightful suggestion. At present, our framework does not support using GO information during inference. This design choice reflects our original intention: the framework is aimed at more challenging real-world virtual screening scenarios, where GO annotations are often unavailable for novel or newly discovered targets. Accordingly, GO is used only during training to enrich the encoder’s representation space, while inference relies solely on sequence or structure input. While incorporating GO during inference could be beneficial when annotations exist, doing so would require a modified multi-entry inference architecture and recalibration of the contrastive embedding space. This goes beyond the current design, but we view it as an interesting direction for future work. We again thank you for the helpful suggestion.
> >
> > ------------------------------------

---

> ### Comment · Reviewer_X8Se · 2025-11-26
>
> Thanks for the additional results. I will keep my positive score.

---

> > ### Author Response · Authors · 2025-11-27
> >
> > Thanks for your prompt and positive feedback.

---

### Meta-Review · Area_Chair_UXvo · 2026-01-07

**Summary:**

Drug-ProGO proposes to leverage Gene Ontology to bridge the gap in protein-level virtual screening when pocket information is absent. The proposed method technically sounds and achieves competitive empirical performance. It receives two positive and two negative scores. Two reviewers noted incremental novelty and modest performance gains. Rebuttal adds analyses but doesn't fully resolve novelty or performance concerns. Considering the integration of GO is the main contribution of the proposed method but the performance benefit of adding this module is not significant, it is recommended as borderline reject.

**Reviewer Concerns:**

Addressed Concerns:
1. The rebuttal addresses concerns of GO evaluation robustness by providing new experiments with GO-similarity filtering, showing consistent improvements over variants without GO.
2. The authors provide results comparing partial fine-tuning to full fine-tuning, showing full fine-tuning does not outperform the current strategy.
3. The authors clarify the usage of AlphaFold structure.
4. New experiments is added to show learnable fusion underperforms the fusion mechanism used in the proposed method.

Outstanding Concerns:
1. While the authors defended the marginal gains by noting the increased difficulty of whole-protein screening compared to pocket-based screening, the empirical results on non-DUD-E datasets remain close to baselines such as LigUnity.
2. While the authors explain why GO integration is technically non-trivial, the core concern remains that the technical novelty beyond adding GO supervision is limited, with the overall architecture still heavily based on LigUnity.
3. While the authors add visualization showing reduced embedding distances with GO, this interpretability of how the GO data improves the model’s performance remains somehow superficial.

**Reviewer Scores:**

Reviewers X8Se and a8ga explicitly state they will maintain at 6.
Reviewer oF9zf may remain at 4 or increase to 6, depending on whether they find the response to the incremental contribution concern satisfactory.
Reviewer pYPJ is unlikely to increase their score, as most of the outstanding concerns are raised by them.

---

### Decision · Program_Chairs · 2026-01-26

Reject